# In-orbit demonstration of an iodine electric propulsion system

Dmytro Rafalskyi[1], Javier Martínez Martínez[1], Lui Habl[1,2], Elena Zorzoli Rossi[1], Plamen Proynov[1], Antoine Boré[1], Thomas Baret[1], Antoine Poyet[1], Trevor Lafleur[1✉], Stanislav Dudin[1] & Ane Aanesland[1]

Propulsion is a critical subsystem of many spacecraft[1–4]. For efficient propellant usage, electric propulsion systems based on the electrostatic acceleration of ions formed during electron impact ionization of a gas are particularly attractive[5,6]. At present, xenon is used almost exclusively as an ionizable propellant for space propulsion[2–5]. However, xenon is rare, it must be stored under high pressure and commercial production is expensive[7–9]. Here we demonstrate a propulsion system that uses iodine propellant and we present in-orbit results of this new technology. Diatomic iodine is stored as a solid and sublimated at low temperatures. A plasma is then produced with a radio-frequency inductive antenna, and we show that the ionization efficiency is enhanced compared with xenon. Both atomic and molecular iodine ions are accelerated by high-voltage grids to generate thrust, and a highly collimated beam can be produced with substantial iodine dissociation. The propulsion system has been successfully operated in space onboard a small satellite with manoeuvres confirmed using satellite tracking data. We anticipate that these results will accelerate the adoption of alternative propellants within the space industry and demonstrate the potential of iodine for a wide range of space missions. For example, iodine enables substantial system miniaturization and simplification, which provides small satellites and satellite constellations with new capabilities for deployment, collision avoidance, end-of-life disposal and space exploration[10–14].

Spacecraft require propulsion to perform manoeuvres in space, such as orbit transfers, avoidance of collisions, orbit maintenance to compensate for aerodynamic or gravitational perturbations, and end-of-life disposal[1]. The choice of propulsion technology, in particular its exhaust speed, determines the propellant mass needed. Electric propulsion[5,15] uses electric power to accelerate a propellant (via electric and/or magnetic fields) and can achieve exhaust speeds that are an order of magnitude higher than chemical propulsion (which uses energy from chemical reactions for propellant acceleration). Some of the most successful electric propulsion systems include gridded ion and Hall thrusters[5], which create a plasma through electron impact ionization of a gas[6] and electrostatically accelerate ions to generate thrust. In addition to being used by many commercial satellites orbiting the Earth, such propulsion systems are also used for space exploration. Examples include the European Space Agency's SMART-1 mission to the Moon[2], NASA's Dawn mission that studied the protoplanets Ceres and Vesta in the asteroid belt between Mars and Jupiter[16], and the Japanese Aerospace Exploration Agency's Hayabusa1 and Hayabusa2 sample-return missions to the near-Earth asteroids 25143 Itokawa[17] and 162173 Ryugu[18].

As spacecraft are power limited, electric propulsion systems must maximize their thrust-to-power ratio, which for electrostatic accelerators requires a propellant with a low ionization threshold and a high atomic mass[5]. At present, the propellant of choice is xenon. However, xenon is very rare (less than one part per ten million in the atmosphere), and commercial production is both expensive and limited[7–9]. There are also competing applications that use xenon, including lighting and imaging, anaesthetics in hospitals[9,19] and etching in the semiconductor industry[20]. With the rise of satellite mega-constellations[21–23], the demand for xenon may outpace supply within the next ten years. A further disadvantage is that xenon must be stored at very high pressures (typically 10–20 MPa), which requires specialized loading equipment and trained personnel, making it incompatible with the 'new space' paradigm. For the long-term sustainability of the space industry, it is critical that a replacement propellant be found.

A possible alternative is iodine[24,25], which is much more abundant and cheaper than xenon[26] (Methods) and can be stored unpressurized as a solid. In addition, both atomic and diatomic iodine have a lower ionization threshold, and diatomic iodine has a relative mass that is almost twice that of xenon. Although iodine is viewed as a game-changing propellant and has been investigated by companies[27,28], universities[29–31] and space agencies[32] around the world, no system has previously been tested in space. Here we describe the development and testing of an iodine electric propulsion system (the NPT30-I2, with a nominal power and thrust of 55 W and 0.8 mN, respectively) and present results of the in-space operation of this new technology.

[1]ThrustMe, Verrières-le-Buisson, France. [2]Laboratoire de Physique des Plasmas, CNRS, Ecole Polytechnique, Sorbonne Université, Université Paris-Saclay, IP Paris, Route de Saclay, Palaiseau, France. ✉e-mail: trevor.lafleur@thrustme.fr

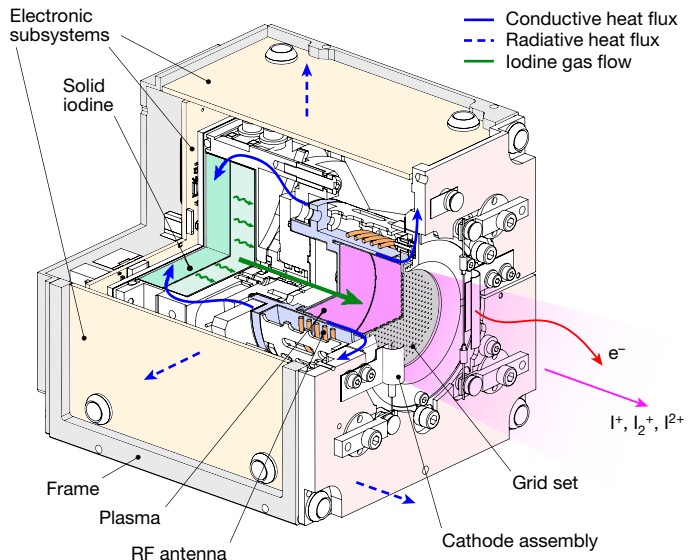

**Fig. 1 | Schematic of the NPT30-I2 iodine electric propulsion system.** Solid iodine (darker green region) is located in a storage tank upstream of the plasma source tube (blue region). Heating causes sublimation and a low-pressure gas (lighter green region) enters the source tube (green arrow). A plasma (purple region) is created by a RF antenna, and iodine ions ($I^+$, $I_2^+$ and $I^{2+}$) are accelerated by a set of grids. A cathode emits electrons ($e^-$) to neutralize the ion beam. Waste heat is conducted towards the iodine tank and structural frame (solid blue arrows) or radiated away (blue dashed arrows).

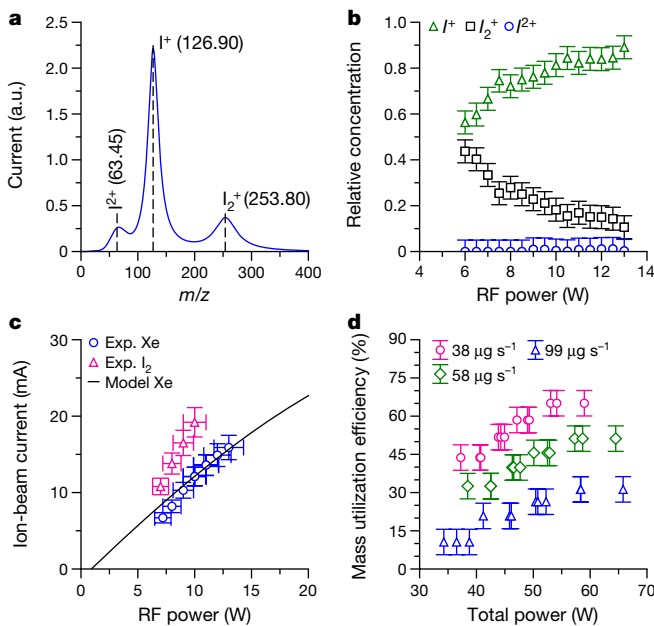

**Fig. 2 | Beam composition and ionization efficiency. a**, Example mass-to-charge ratio, $m/z$, spectrum obtained with the TOF diagnostic system. The labels indicate the ions $I^{2+}$, $I^+$ and $I_2^+$. **b**, Relative current concentration of iodine species in the ion beam as a function of RF generator output power. **c**, Ion-beam current extracted from the plasma source as a function of RF power with iodine and xenon propellants. The black curve shows results of a numerical plasma discharge model (Methods). **d**, Propellant mass utilization efficiency as a function of total power for different iodine mass flow rates. The error bars represent estimates of measuring equipment precision and accuracy limitations.

Solid diatomic iodine is stored in a tank connected to an inductively coupled plasma source tube terminated by two high-voltage, multi-aperture grids (Fig. 1). Heaters connected to the tank cause iodine sublimation and subsequent gas flow into the source tube. An iodine plasma is created by electron impact ionization using a radio-frequency (RF) inductive antenna, and positive plasma ions are extracted and accelerated by the grids to high speeds (about 40 km s$^{-1}$) to produce thrust. A cathode filament downstream of the grids thermionically emits electrons to charge neutralize the ion beam. The propulsion system includes all subsystems for operation (Extended Data Fig. 1a), such as propellant storage, delivery and control, the gridded ion thruster, electron-emitting neutralizers, the power processing unit and passive thermal management[33]. See Methods for further details of the electrical system and cathode neutralizer. Iodine enables substantial miniaturization, and with the innovations discussed below, the total mass (including propellant) and volume are 1.2 kg and 96 mm × 96 mm × 106 mm, respectively.

The use of iodine creates unique design and operational challenges. Iodine has a high electronegativity that can lead to corrosion with many common materials. Technical ceramics (aluminium oxide and zirconium oxide) are used for the source tube and some interface components, and all vulnerable metal surfaces are coated with a polymer film. The iodine sublimation rate is controlled by monitoring and adjusting the tank temperature to maintain the desired saturation pressure in the range of 2–6 kPa. The operating temperature of the tank is kept between 80 °C and 100 °C to avoid local melting of iodine, and the tank is directly integrated upstream of the plasma source tube. When the propulsion system is not firing, iodine gas cools and deposits within a small orifice (Methods) between the tank and source tube blocking further flow without the need for a control valve.

Vibrations during launch and spacecraft motion once in orbit can cause solid iodine to break into pieces, which may damage the propulsion system or lead to poor thermal contact during heating. To prevent this, iodine is embedded into a porous aluminium oxide ceramic block

with a porosity of 95% placed inside the tank (the tank-to-propellant mass fraction is 54%). During assembly, iodine is heated above its melting temperature to form a liquid, which is poured into the block (Methods). Once cooled, the iodine solidifies and is safely held. When the propulsion system fires, plasma heat losses to the walls of the source tube and heat losses in the power electronics are directed towards the storage tank (Fig. 1). This allows reuse of waste heat so that less than 1 W of additional power is needed from heaters during steady-state operation. All other heat losses are channelled to the front and side panels of the propulsion system and either radiated or conducted to the spacecraft. See Methods for further thermal design details.

For plasma creation and ion acceleration, the use of iodine leads to important differences over xenon, as in addition to the molecular ion $I_2^+$, direct dissociative ionization[29] and two-step dissociation and ionization reactions allow the formation of the atomic ion $I^+$. Multiply charged ions, such as $I^{2+}$, are also possible. Ground testing has been conducted to characterize the system before launch (Methods). Using time-of-flight spectrometry with an electrostatic diagnostic system in the thruster plume, mass-to-charge ratio spectra are measured (Fig. 2a) and the beam composition is determined, as shown in Fig. 2b. The dominant ion species are $I_2^+$ and $I^+$, and their relative fractions change with RF generator output power. As the mass flow rate is fixed, higher iodine dissociation occurs at higher powers owing to an increased plasma density. Gas depletion at these powers also results in more energetic electrons, which affects the rate factors of collisional processes[29].

Despite diatomic gases having additional energy-loss mechanisms associated with molecular dissociation and the excitation of vibrational and rotational states[34], the ionization efficiency of iodine in our propulsion system is higher than that of xenon, as shown in Fig. 2c. For the experiments with xenon, the system was temporarily modified to inject gas into the source tube from an external high-pressure storage tank.

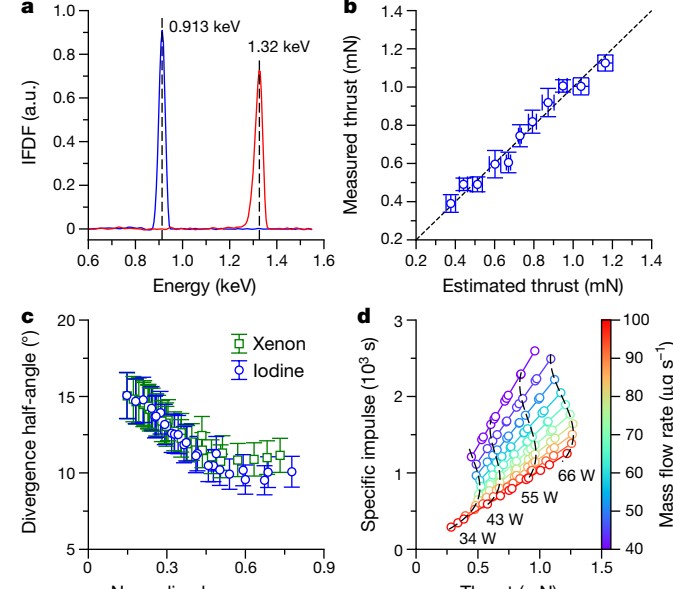

**Fig. 3 | Propulsion system performance. a**, Ion flux distribution functions (IFDF) in the plume for acceleration voltages of 900 V and 1,300 V. **b**, Direct thrust measurements from a thrust balance compared with indirect thrust measurements estimated from the ion-beam current, applied grid voltage, and extrapolated beam divergence and beam composition data. **c**, Measured ion-beam divergence half-angle with iodine and xenon. The normalized perveance, $p/p_{max}$, is a measure of the ion space charge (Methods). **d**, Thrust and specific impulse performance map of the propulsion system within the operating total power range, and for different iodine mass flow rates. The error bars represent 1 s.d. (**b**) or estimates of measuring equipment precision and accuracy limitations (**c**).

The iodine mass flow rate is inferred from measurements of the entire propulsion system mass before and after operation. In Fig. 2c, for the same mass flow rate and RF power, an almost 50%-higher beam current is extracted from the plasma source with iodine. This improvement is consistent with previous experimental and numerical results[29,30] and occurs because of the lower ionization threshold of iodine ions (10.5 eV for $I^+$ and 9.3 eV for $I_2^+$) compared with xenon (12.1 eV for $Xe^+$) and the different collisional processes and reaction cross-sections. This results in a lower electron temperature and lower consequent plasma losses to the source tube walls[34]. The results for xenon are in agreement with a numerical model (Methods). A common ionization performance metric[5,35] is the propellant mass utilization efficiency (Fig. 2d), $\eta_m = \dot{m}_i/\dot{m}$, where $\dot{m}_i$ is the ion mass flow rate and $\dot{m}$ is the sublimation mass flow rate. At the highest performance in our system, $\eta_m \approx 60\%$ for iodine and $\eta_m \approx 40\%$ for xenon (not shown).

Ions in the plasma source are extracted and accelerated by voltages between 800 V and 1,300 V applied across the grids. Measurements of the ion flux distributions (Methods) in Fig. 3a confirm the presence of high-energy ions with an average energy close to the net accelerating voltage, $V_n$, of 900 V and 1,300 V, respectively. By measuring the ion-beam current (Methods) an indirect measurement of the thrust is obtained from, $F = \alpha \gamma I_{beam}\sqrt{2M_I V_n/q_I}$. Here $I_{beam}$ is the beam current, $M_I$ and $q_I$ are the mass and charge of atomic iodine ions, respectively, and $\gamma$ and $\alpha$ are correction factors: $\gamma = \cos\theta_{div}$, where $\theta_{div}$ is the beam divergence half-angle, and $\alpha = \beta_{I^+} + \sqrt{2}\beta_{I_2^+} + \beta_{I^{2+}}/\sqrt{2}$, where $\beta_{I^+}$, $\beta_{I_2^+}$ and $\beta_{I^{2+}}$ are the relative current contributions for each ion species and the pre-factors represent the square root of the relative mass-to-charge ratio. An example of the thrust correction factor, $\alpha\gamma$, is shown in Extended Data Fig. 3b. The propulsion system electronics continually perform these indirect thrust estimates during operation. Direct thrust measurements are obtained using a thrust balance (Methods). Figure 3b shows the measured thrust range achievable, and a comparison between direct and indirect measurements.

By carefully designing the grids (Methods, Extended Data Fig. 3a, d), ions are well focused with a low divergence between 10° and 15°, as shown in Fig. 3c. The beam divergence has been measured with an automated array of electrostatic probes (Methods). The iodine divergence is slightly lower than that of xenon because of the improved ionization efficiency, which reduces unionized neutrals in the plume and lowers the ion-neutral collision frequency. An important performance metric is the specific impulse[5], $I_{sp} = F/\dot{m}g_0$, which represents how effectively propellant is used (here $g_0$ is the gravitational acceleration equal to 9.81 m s$^{-2}$). The performance map of the propulsion system is shown in Fig. 3d (see also Extended Data Fig. 3c, e), where the maximum thrust and specific impulse are about 1.3 mN and 2,500 s, respectively, for total powers (which includes RF power, acceleration power, neutralizer power, propellant heating power, electronics power and all losses) below 65 W. The total impulse that can be delivered to a spacecraft at the maximum specific impulse is 5,500 Ns (corresponding to a burn time of about 1,500 h).

The propulsion system has undergone extensive qualification to meet in-space conditions and launch-vehicle requirements (Methods), and a flight model was recently integrated into the Beihangkongshi-1 satellite operated by Spacety (Extended Data Fig. 1b). The 12-unit CubeSat (with a mass of approximately 20 kg) was launched into space onboard a Long March 6 rocket on 6 November 2020. The satellite was injected into a circular, Sun-synchronous orbit with an altitude of approximately 480 km.

Figure 4a summarizes all test firings performed up until 28 February 2021 and shows the mean semi-major axis of the satellite as predicted from a theoretical model, GPS data from the satellite, numerical simulations (using the General Mission Analysis Tool, GMAT[36]), and independent tracking data of the satellite (satellite catalogue number 46838) produced by the Space Surveillance Network (SSN) operated by the US Space Command[37]. The arrows indicate 11 firings over the displayed time period. Tests 1A and 1B represent firings to check the overall system operation. Subsequent firings 2A–2I test the repeatability and ignition cycling. The direction of the thrust vector has been varied during some firings (by reorienting the satellite using its onboard attitude control system). The duration of each test is between 80 min and 90 min (including 10–20 min for iodine heating and plasma ignition, which results in a small propellant mass loss of 12 mg before thrust generation), which for each firing gives an altitude change between 200 m and 400 m at a thrust and power of about 0.8 mN and 55 W, respectively. As an example, Fig. 4b shows GPS data, GMAT simulations and theoretical predictions for firing 1B, and Fig. 4c shows the estimated thrust and power consumption from telemetry data during the manoeuvre (see also Extended Data Fig. 4c). See Methods and Extended Data Table 1 for further orbital analysis details.

The results in Fig. 4a, b shows definite orbit changes correlated with the known propulsion system start times. At present, the firings have demonstrated a cumulative altitude change above 3 km. Additional downloaded propulsion system telemetry data are shown in Fig. 4d (see also Extended Data Fig. 4a, b), which is compared with corresponding ground measurement data taken during qualification. These measurements confirm that sufficient ion beam neutralization occurs (the electron emission current is larger than the ion current) and that ground testing conditions reproduce the space environment.

The linear decay between firings in Fig. 4a represents residual aerodynamic drag on the satellite[1]. Manoeuvres 1A and 1B demonstrate that the propulsion system can be used for orbit maintenance to compensate for this drag. In addition, all firings are representative of collision avoidance manoeuvres. Given the rapid growth of small satellites in low Earth orbit[38], a miniaturized propulsion system enabled by the use of iodine will provide such satellites with the capability to avoid potential collisions and to deorbit at end of life to prevent the build-up of space

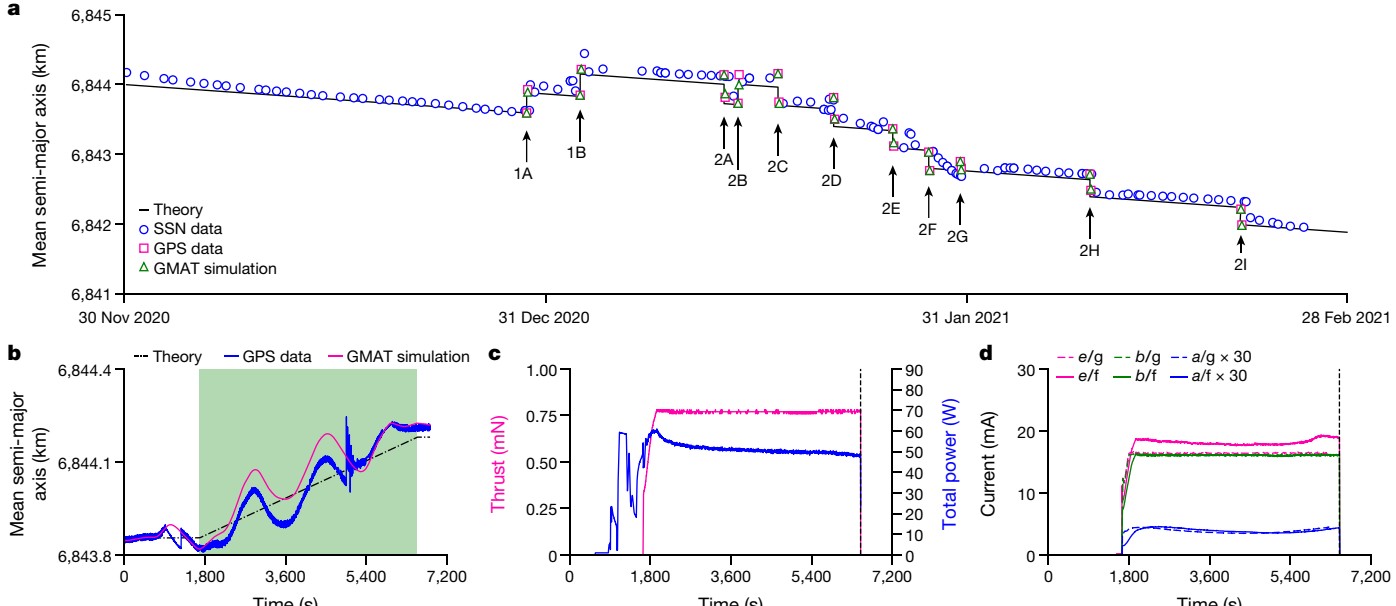

**Fig. 4 | In-orbit manoeuvres performed by an iodine electric propulsion system. a**, Mean semi-major axis of the Beihangkongshi-1 satellite from the SSN[38] and GPS data, and as predicted using numerical simulations and theory. The arrows indicate separate firings. **b**, Mean semi-major axis as a function of time during manoeuvre 1B. The green region indicates when the propulsion system is firing. **c**, Thrust and total power telemetry during manoeuvre 1B. **d**, Comparison between ion-beam current, *b*, electron neutralizer current, *e*, and current to the accel grid, *a*, during ground, *g*, and in-flight, *f*, operation for manoeuvre 1B. The GPS data have an accuracy of approximately 20 m.

debris: actions that will prove vital for the long-term sustainability of the space industry[39].

In conclusion, we have described an iodine electric propulsion system and presented in-orbit results demonstrating this new technology. Our work shows that iodine is not only a viable replacement propellant for xenon but also gives enhanced performance. For large satellites and satellite constellations, the use of a more abundant propellant that can be stored unpressurized will help simplify satellite design and propulsion system integration and reduce the market demand for xenon, which may have benefits in other sectors[9,24]. For smaller satellites, iodine provides high impulse capability giving new options for deployment, collision avoidance and deorbiting, and advanced space exploration missions[10–14].

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

## Methods

### Propellant cost comparison

High-purity iodine is not needed in our propulsion system, and the total propellant cost for a purity of 99.5% was approximately US$60, with an additional cost below US$200 for iodine-related hardware. The propulsion system qualification cost was just under US$4,000. A modified version of our propulsion system using xenon propellant has also been developed. For the same mass of propellant, the cost of xenon was US$1,275, and owing to the high-pressure titanium tank, flow control valves, pipe and sensors, the hardware cost was about 100 times higher than that for iodine. The qualification cost also increased to approximately US$9,000.

The high cost of xenon is one of the reasons that SpaceX has instead chosen krypton as an alternative propellant for their Starlink satellites[40]. However, krypton has a higher ionization threshold and lower atomic mass than both xenon and iodine, and the required propulsion system power increases by more than 25% to achieve the same thrust level. Furthermore, the storage density of krypton is approximately three times lower than that of xenon (and nine times lower than that of iodine)[41], which increases the volume and mass of the propellant tank. Krypton is used in a number of competing industries, such as window insulation applications, which account for more than 50% of the market share, and which is expected to grow rapidly[42] due to the demand for energy-efficient buildings. Considering a Starlink constellation size between 12,000 and 42,000 satellites, each requiring of the order of 10 kg of propellant[40], a substantial amount of krypton will be needed in the coming years.

### Plasma modelling

Owing to the similarity in physical properties between xenon and iodine, and the wider availability and accuracy of important physical data (such as reaction cross-sections), xenon has been used in numerical modelling to aid the design, development and testing of the propulsion system. The model (which is similar to that in ref. [43]) is self-consistent and considers mass flow conservation, and volume-averaged ionization and power balance within the thruster summarized by the following steady-state conservation equations

$$\dot{m} = \dot{m}_i + \dot{m}_n \tag{1}$$

$$A_{eff}\, u_B = n_n K_{iz} V \tag{2}$$

$$P_{RF} = \frac{n_p u_B A_{eff} \varepsilon_T}{\eta_{rf}} \tag{3}$$

Here $\dot{m}$ is the total input propellant mass flow rate, $\dot{m}_i$ is the mass flow rate of ions extracted from the plasma source, $\dot{m}_n$ is the mass flow rate of any unionized neutral gas, $A_{eff}$ is the effective surface area for plasma loss inside the source tube, $u_B = \sqrt{qT_e/M}$ is the Bohm velocity[34], $q$ and $M$ are the ion charge and mass, respectively, $T_e$ is the electron temperature, $n_n$ is the average neutral gas density in the source tube, $K_{iz}$ is the ionization rate factor, $V$ is the volume of the source tube, $P_{RF}$ is the RF power, $n_p$ is the average plasma density inside the plasma source, $\varepsilon_T$ is the effective energy loss per ion-electron pair lost to the source tube walls[34,43] (and which includes both collisional and kinetic energy losses) and $\eta_{rf}$ is the antenna–plasma power transfer efficiency. The equations above implicitly include relevant electron-neutral reaction processes (such as elastic scattering, and inelastic excitation and ionization processes), RF antenna–plasma coupling and plasma-wall sheath physics[34,43].

### Ion optics

One of the key elements of a gridded ion thruster are the ion optics, which in our case consists of a two-grid assembly. Particle-in-cell (PIC) simulations (using the open-source code XOOPIC[44]) have been used to model ion extraction and acceleration by the grids. Extended Data Fig. 3d shows a PIC simulation of ion acceleration through a single set of grid holes. Ions are well focused through the holes with no direct impingement, and only low-energy ions generated by ion-neutral charge-exchange collisions and possible downstream electron-neutral ionization with unionized neutral gas in the plume, strike the second grid (called the accel grid), as is typical in gridded ion thrusters[5].

For given grid dimensions, if the total accelerating voltage is too low, the space charge of the ions between the grids can lead to under-focusing and direct impingement on the upstream surface of the second grid. This results in rapid sputter erosion, and possible shorting of the grids by sputtered material. The well known Child–Langmuir law[34] can be used to estimate this space-charge-limited current, $I_{CL}$, which in our case gives

$$I_{CL} = \frac{4\varepsilon_0 N A_s}{9} \sqrt{\frac{2q}{M}} \frac{V_T^{3/2}}{L_{eff}^2} \tag{4}$$

where $\varepsilon_0$ is the permittivity of free space, $N$ is the number of grid apertures, $A_s$ is the area of each aperture in the upstream grid (called the screen grid), $V_T$ is the total accelerating voltage across the grids and $L_{eff} = \sqrt{(t_s + L_g)^2 + r_s^2}$ is the effective grid gap length with $t_s$ and $r_s$ the screen grid thickness and aperture radius, respectively, and $L_g$ the physical grid gap length. A useful metric for quantifying the level of space charge between the grids is the perveance, $p = I_{beam}/V_T^{3/2}$. When the ion beam current is equal to the space-charge-limited current, the maximum perveance, $p_{max}$, of the grids is reached. For the grids used in our propulsion system, $p_{max} = 1.7 \times 10^{-6}$ A V$^{-3/2}$ for singly charged atomic iodine ions. If the total accelerating voltage is instead too high, the cross-over limit[5] is reached and ions are over-focused, again leading to erosion. The space-charge and cross-over limits are indicated in Extended Data Fig. 3a.

### Thermionic cathode neutralizer

Conventional electric propulsion systems typically make use of hollow cathode plasma bridge neutralizers[5], which are capable of emitting a high electron current and are well suited to neutralizing large ion-beam currents. As our propulsion system operates at low power, and to further enable system miniaturization, two thermionic carburized thoriated tungsten filament neutralizers are used instead with a total estimated lifetime of 3,600 h.

### Electrical system design

The electronics system is separated into modules as shown in Extended Data Fig. 2a. A main control unit coordinates the operation of the propulsion system, whereas each of the other modules controls a functional component by providing local regulation and monitoring of relevant parameters. The propulsion system is supplied by an unregulated voltage bus in the range 10–30 V and requires a power between 30 W and 70 W depending on the operating mode. A common mode filter on the power line reduces electromagnetic interference. The main communication channel with the satellite is a redundant Controller Area Network bus operating at data rates between 250 kbit and 1 Mbit. In addition, an inter-integrated circuit interface can also be used. Galvanic isolation is implemented on all communication channels.

The propulsion system uses five microcontrollers: one main processor and four second-level controllers managing local subsystems. The main microcontroller implements global control and safety algorithms, and also provides the interface with the satellite's onboard computer (OBC). A real-time operating system with multiple tasks is used, where each task has a priority assigned and the scheduler decides which should be executed depending on the given priority.

After receiving a firing request from the OBC, the propulsion system switches on the subsystems, carries out built-in self-tests and proceeds with the plasma ignition sequence. Each microcontroller implements

a bootloader allowing the OBC to reprogramme the user application in flight. This bootloader has several safety measures, such as redundancy or a triple voting algorithm, to avoid possible corruption caused by single-event upsets.

### Thermal design

Heat is generated by ohmic losses in the power electronics and plasma losses to the source tube walls. The internal components of the gridded ion thruster reach the highest temperatures (up to 170 °C), whereas all other components and subassemblies are below 80 °C. The amount of heat needed for iodine sublimation is given by

$$Q = \dot{m}_{I2} \Delta H_s \tag{7}$$

where $\dot{m}_{I2}$ is the mass flow rate and $\Delta H_s$ is the sublimation enthalpy of iodine (62.4 KJ mol$^{-1}$). For a typical mass flow rate of 0.07 mg s$^{-1}$, the sublimation power is less than 0.02 W. Owing to the reuse of waste heat, less than 1 W of additional power is needed by the flow management system to compensate for conductive and radiative losses and keep the propellant flow path sufficiently hot to prevent iodine deposition. Both the tank and flow path to the source tube have heaters maintaining the target temperature during start-up, ignition and steady-state operation. For a cold start, approximately 10 min is needed to heat the iodine to the required temperature.

### Propellant valve

To enhance miniaturization and eliminate moving parts, the propulsion system does not use a conventional solenoid control valve. Instead, controlled iodine deposition and blocking of a submillimetre hole between the propellant tank and source tube is used. When the propulsion system is not operating, the temperature of the orifice causes deposition, which blocks the hole. At this deposition temperature, the resulting sublimation rate is very low. In addition, the effective gas flow conductance is substantially reduced owing to the design of the orifice, the gas distribution head, source tube and acceleration grid themselves, so that iodine leakage is low. Ground-based experiments with the propulsion system stored under vacuum for over two weeks show an upper limit leakage rate of less than 0.08 μg s$^{-1}$.

### Propellant loading

Iodine is filled into the porous matrix, which is placed inside the propellant storage tank before the filling process. To improve thermal conductivity, a polymeric thermal pad is placed between the matrix and the walls of the tank. Although iodine does not have a strong chemical affinity with oxygen under normal conditions, owing to its oxidizing nature the tanks are purged with argon before propellant filling to remove any residual gases that could contaminate the plasma during operation.

Iodine is melted at a temperature close to 120 °C in a separate reservoir and immediately poured into the matrix. This improves the packing factor over typical solid iodine crystals, and helps to minimize the formation of voids. The absolute pressure in the reservoir is just above atmospheric pressure with the argon partial pressure kept close to 100 kPa. A saturated state is maintained inside the tank as the vapour pressure of iodine is close to 14 kPa at 120 °C (ref. [45]).

### Diagnostics for ground testing

#### Vacuum chamber testing. 
Performance and plume characterization was performed by operating the propulsion system inside a cylindrical vacuum chamber with a length of 0.83 m and a diameter of 0.6 m. The chamber was pumped with a combination of rotary, turbo-molecular and cryogenic (operated at −75 °C) pumps. The pressure was measured with a MKS Baratron 627B absolute pressure transducer and a cold cathode Balzers IKR 050 gauge (with gas-specific correction factors applied). The chamber base pressure was better than $5 \times 10^{-4}$ Pa, with a background pressure below $1.4 \times 10^{-3}$ Pa maintained during operation. Although the neutral iodine gas dissociation fraction is not well known in the chamber, the effective pumping speed is estimated to be between 700 l s$^{-1}$ and 1,400 l s$^{-1}$.

**Automated beam diagnostic system.** Ion-beam current and divergence measurements are performed with a semi-circular, automated, beam diagnostic system[46] that includes an array of 15 planar electrostatic probes. Motors at each end of the semi-circular arm precisely control the azimuthal arm position, which allows spatial measurements of the ion-beam current density over a two-dimensional hemispherical surface. The probes are biased at −40 V to reflect electrons and any possible negative iodine ions in the plume. The measured current is corrected to account for secondary electron emission due to ion bombardment of the probes and plasma sheath expansion around each probe due to the applied voltage[46]. The total ion-beam current, $I_{beam}$, and effective beam divergence half-angle, $\theta_{div}$, are obtained by integrating the measured current density profiles according to the following equations

$$I_{beam} = R^2 \int_{-\frac{\pi}{2}}^{\frac{\pi}{2}} d\Phi \int_{-\frac{\pi}{2}}^{\frac{\pi}{2}} d\theta J_i(\Phi, \theta) \tag{9}$$

$$\theta_{div} = \cos^{-1} \left[ \frac{\int_{-\frac{\pi}{2}}^{\frac{\pi}{2}} d\Phi \int_{-\frac{\pi}{2}}^{\frac{\pi}{2}} d\theta J_i(\Phi, \theta) \cos\Phi \cos^2\theta}{\int_{-\frac{\pi}{2}}^{\frac{\pi}{2}} d\Phi \int_{-\frac{\pi}{2}}^{\frac{\pi}{2}} d\theta J_i(\Phi, \theta)} \right] \tag{10}$$

where $R$ is the radius of the semi-circular probe arm, $\Phi$ and $\theta$ are the probe azimuthal and latitude angles, respectively, and $J_i$ is the ion beam current density.

**Time-of-flight probe.** Time-of-flight (TOF) measurements were performed using a molybdenum disk with a diameter of 7 cm placed in the plume, and located 54 cm downstream of the accel grid. The probe was biased at −100 V to reflect electrons and any possible negative iodine ions[29] in the plume, and the current collected by the probe was measured with a digital oscilloscope using short, low-impedance, connections. The time constant of the probe is much less than the ion transit time and is of the order of 1 μs. During measurements, both grids of the propulsion system are initially grounded before a rectangular voltage pulse with an amplitude and width of 1,000 V and 4.5 μs, respectively, is applied (with rising and falling times of approximately 0.5 μs). This causes an instantaneous extraction and acceleration of positive ions from the plasma source, and the appearance of distinct peaks in the measured TOF probe current due to the different ion transit times, $\tau$, to the probe

$$\tau = \frac{L}{\sqrt{2qV_n/M}} \tag{11}$$

where $L$ is the distance between the exit of the propulsion system and the TOF probe, $V_n$ is the net accelerating voltage and $q/M$ is the ion charge-to-mass ratio. Owing to pulse-shape limitations, probe current peaks show a certain spread. Individual ion species contributions are determined by fitting exponential Gaussian functions and integrating to find the average current.

**Retarding field energy analyser.** A Semion 2500 Retarding Field Energy Analyzer (RFEA) from Impedans is used to measure the distribution function of beam ions. The RFEA has a diameter of 50 mm and includes a single grounded front grid, two internal grids with a controlled bias voltage and a biased collector plate. The RFEA is located 30 cm downstream of the propulsion system and is connected to an automated Semion Electronics Unit scanning system. The first derivative of the collector

current as a function of the swept bias voltage[47], $V_{bias}$, on the second grid then gives the ion flux distribution function, $h(V_{bias})$, defined such that

$$h(V_{bias}) \propto \frac{dI_{RFEA}}{dV_{bias}} \quad (12)$$

where $I_{RFEA}$ is the collector current measured by the RFEA.

**Indirect thrust measurements.** The integrated electronics in the propulsion system includes current and voltage measurement sensors that continually measure the applied accelerating voltage, and the current to both grids. For gridded ion thrusters, the ion-beam current that is extracted from the plasma source is balanced by an electron current to the first grid to maintain charge balance (Extended Data Fig. 2b). This current, after subtracting off the small current from the accel grid, then matches the net electron current emitted by the thermionic cathode neutralizer. During ground testing, and operation in space, the grid current and voltage measurements allow estimates of the extracted beam current and thrust to be made in real time.

**Direct thrust measurements.** Direct thrust measurements were performed with the propulsion system attached to a thrust balance placed inside the vacuum chamber. We developed a single pendulum thrust balance with a sensitivity of 0.03 mN that uses a force sensor to measure the thrust applied at the end of a moving arm. The force sensor and thrust vector location are shifted, which changes the respective pendulum lever arms and allows the measured force on the sensor to be magnified. The force sensor is an S256 load cell with an integral overload stop, which produces an analogue voltage output with a sensitivity of 1 mV V$^{-1}$ at full-scale load (100 mN). To remove electrical interference, the low-level output voltage from the load cell is converted to a digital signal and sent to the measuring unit located outside of the chamber. The raw data are digitally smoothed with a second-order Savitzky–Golay filter. The thrust balance is calibrated with a set of known masses placed on a horizontal arm that produces a moment about the pendulum pivot balanced by the moment due to the force on the sensor.

## Diagnostics for in-flight testing

The propulsion system includes eight thin-film platinum temperature sensors for measurement of the temperature (with an accuracy of 0.1 °C) at key locations, including all electronic subsystems, the propellant tank, and the interface flange between the tank and plasma source tube. The input current and voltage from the satellite, as well as output currents and voltages from different subsystems, such as the cathode neutralizer, grids and RF antenna, are continuously measured. The data acquisition frequency is set by the satellite onboard computer and is equal to 1 Hz.

## Flight qualification

The propulsion system has undergone extensive vibration, radiation, thermal and flow testing for flight qualification. Vibration testing consisted of sinusoidal, random and sine-burst (quasi-static acceleration) testing at levels set by the spacecraft launch vehicle. Sinusoidal vibrations include low-frequency tests (5–100 Hz) with accelerations up to 4.5 g. Random vibration tests ranged between 20 Hz and 2,000 Hz with a total root-mean-square acceleration of 6.7 g and a duration of 120 s per axis. Quasi-static tests were also performed for each axis with a maximum acceleration of 8.75 g. Additional shock tests were conducted at frequencies up to 5,000 Hz, with a shock response spectrum acceleration up to 1,500 g. Electronic components and electromechanical assemblies underwent single-event radiation testing (high-energy proton bombardment) at energies up to 200 MeV, as well as gamma-ray testing for a total ionizing dose compatible with a qualification level of 15 krad for the unshielded assembly. The entire propulsion system underwent thermal exposure and thermal cycling campaigns in both ambient conditions and under vacuum conditions in a thermal vacuum chamber (with temperatures between −25 °C and 60 °C). Propulsion system operation in a vacuum chamber confirmed iodine sublimation and overall performance stability over extended firing times with multiple on–off cycles. Long-term propulsion system operation was tested with a qualification model for a total cumulative time of 120 h with 109 separate on–off ignition cycles.

## Collection and analysis of in-flight data

The propulsion system electronics records approximately 50 telemetry parameters that are downloaded from the satellite after each in-orbit firing. The thrust and power depend on the operational mode selected, with different modes possible depending on the power, mass flow rate and applied grid voltage. Two modes have been tested during the in-orbit demonstration as shown in Extended Data Table 1, and denoted N1 and FS. The N1 mode has a thrust-locked feedback loop with a target thrust of 0.8 mN and an upper limit of 60 W, whereas the FS mode has a minimum thrust of 0.35 mN with an upper power limit of 50 W. In this last mode, the propulsion system has all feedback loops disabled, and data from secondary sensors are ignored. An automated self-test is performed before each firing.

Example system temperature measurements performed during in-orbit operation are compared with ground testing measurements in Extended Data Fig. 4b. The results are similar for all parameters, and again show that ground testing conditions replicate the space environment.

Orbit changes resulting from each firing were confirmed using both direct and indirect evidence. Direct evidence includes satellite tracking data from a GPS receiver onboard the satellite, and independent tracking data obtained from the SSN (see ref. [37] with satellite catalogue number 46838). Indirect evidence comes from a comparison of satellite orbital elements calculated from the GPS data with those predicted by numerical simulations using GMAT[36], and a simplified theoretical model based on low-thrust trajectories around a spherical Earth[48]. The theoretical model uses the GPS mean semi-major axis just before manoeuvre 1A begins as an initial condition (backpropagating for earlier times). GMAT simulations use the JGM-3 geopotential model of degree and order 70 × 70, as well as point mass perturbations for the Moon and Sun. Atmospheric drag is included using the MSISE90 model[49], as well as solar radiation pressure using a spherical spacecraft model[36]. Simulations are initiated using times and positions from the GPS data before each firing begins, and use approximate thrust profiles taken from the downloaded telemetry.

Owing to gravitational perturbations, the osculating semi-major axis of the satellite shows oscillations with an amplitude of the order of 10 km. For this reason, mean orbital elements based on Brouwer theory[50] are used, which smooth out these high-frequency oscillations. The mean semi-major axis is deduced from the SSN data after converting from the Kozai to Brouwer mean motion convention[51].

## Data availability

The raw in-orbit data generated and/or analysed during this study are not publicly available as they are partially owned by ESA and Spacety, but are available from the corresponding author on reasonable request and with permission from ESA or Spacety. The SSN data in Fig. 4a that support the findings of this study are available from www.space-track.org (satellite catalogue number 46838). Source data are provided with this paper.

## Code availability

The codes that support the findings of this study are available from the corresponding author upon reasonable request.

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

**Acknowledgements** We thank the European Space Agency (ESA) for supporting the in-orbit demonstration through the TURBO project, ARTES C&G programme, contract number 4000131883/20/NL/MM/can, the French Government (Ministère de l'Enseignement Supérieur, de la Recherche et de l'Innovation) for supporting this work through the i-Lab 2017 grant (le Grand Prix I-LAB 2017 – 19th Concours Nationale d'Aide à la Création d'Entreprises des Technologies Innovantes), and the French National Space Agency (Centre National d'Etudes Spatiales) for support of the project through the INODIN grant, R&T action R-S19/PF-0002-108-92. We thank Spacety, and particularly J. Zheng, for mission support during the in-orbit demonstration. We also thank the Laboratoire de Physique des Plasmas at Ecole Polytechnique for funding open access of our paper.

**Author contributions** D.R. conceived the original prototype and operational algorithms, and, together with A.A., elaborated the concept and development roadmap. T.B. and A.P. performed mechanical design and mechanical simulations of the system. J.M.M. performed thermal and fluid dynamics characterization of the system, and characterization of the chemical interaction of iodine with system components. D.R., E.Z.R., L.H. and A.B. performed experimental plasma and ion beam characterization and supported in-orbit testing and thruster telemetry data analysis. P.P., together with A.B., designed and developed the hardware and system software. S.D. developed the RF generation system, and, together with E.Z.R., contributed to the direct thrust measurements. L.H. and T.L. performed ion optics and plasma modelling. T.L. and J.M.M. performed orbital dynamics simulations and analysis of in-flight telemetry and GPS data obtained during the mission. D.R., T.L., L.H. and J.M.M. planned and coordinated the writing of the manuscript. All authors contributed to the design process and to the editing of the manuscript.

**Competing interests** D.R., J.M.M., L.H., E.Z.R., P.P., A.B., T.B., A.P., T.L. and A.A. are employees of ThrustMe. S.D. is a consultant working with ThrustMe. D.R. and A.A. hold a patent related to the propulsion system (patent no. WO2017037062A1).

**Additional information**
**Correspondence and requests for materials** should be addressed to Trevor Lafleur.

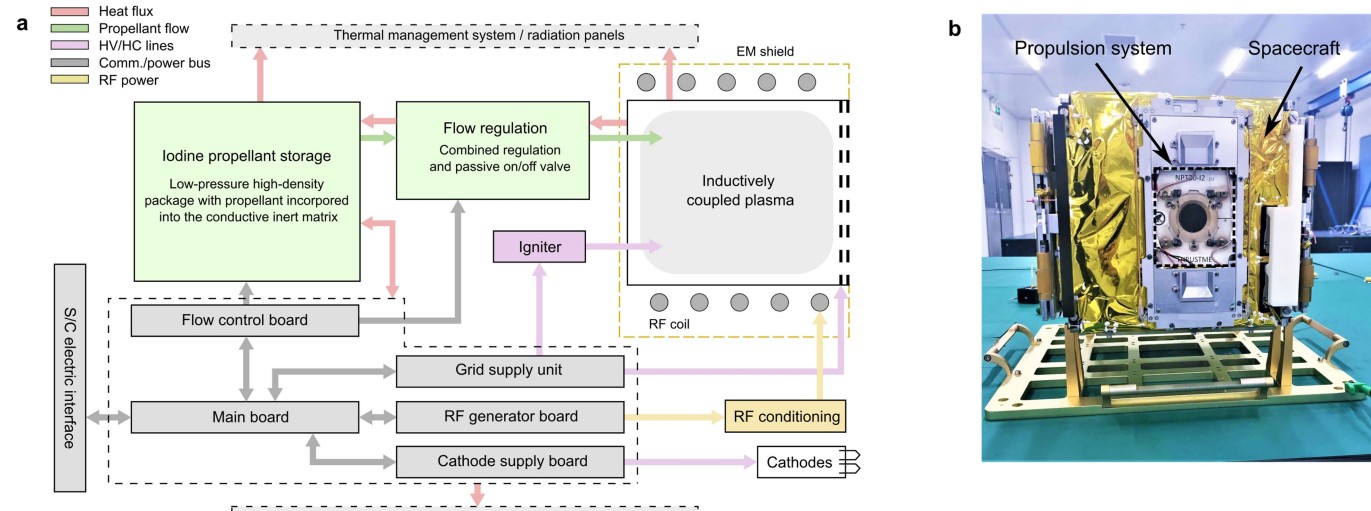

**Extended Data Fig. 1 | Propulsion system architecture and integration with satellite. a**, The propulsion system is a complete system that includes all necessary subsystems for operation. Power is supplied from the spacecraft (S/C) and used for flow control, plasma generation, ion acceleration, and beam neutralization. Solid iodine sublimates and enters the inductively coupled plasma source. An igniter initially strikes a plasma which is then maintained by an RF antenna wrapped around the outside of the source tube. Ions from the plasma are extracted and accelerated by the high-voltage grids, and the positive ion beam is neutralized by electrons thermionically emitted from the cathode filament. **b**, The propulsion system installed in the Beihangkongshi-1 satellite before launch. Photograph reproduced and adapted by the authors with permission from Spacety. © 2020 Spacety Co., Ltd. (Changsha).

**a**

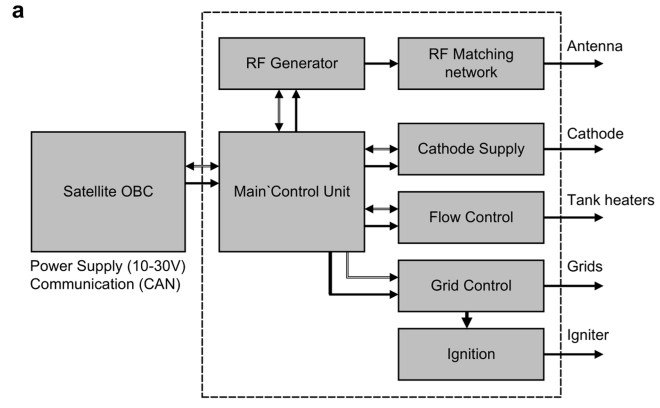

**b**

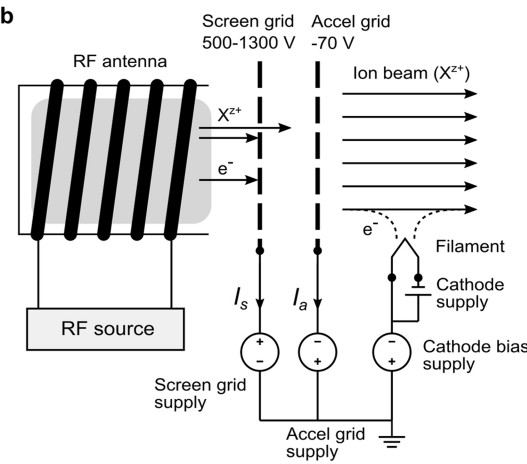

**Extended Data Fig. 2 | Electrical system architecture and thruster electrical schematic. a**, The main control unit interfaces with the satellite onboard computer and implements global control and safety algorithms. The RF generator supplies power to the RF antenna via a matching network to match the impedance of the plasma and generator for efficient power transfer. The cathode supply controls and monitors the electron-emitting cathode filament, the flow control unit manages the propellant tank and flow path heaters, the grid control unit manages the applied voltage to the acceleration grids, and the ignition unit controls the igniter needed for initial gas breakdown in the source tube to produce a plasma. **b**, General electrical circuit showing the high-voltage grids and electron-emitting cathode. Ions (denoted $X^{z+}$) from the upstream plasma source are extracted and accelerated by the voltage applied across the screen and accel grids. A small ion current, $I_a$, flows to the accel grid due to charge-exchange collisions with any unionized propellant in the plume. To maintain charge balance in the source tube, an electron current equal to the sum of the ion beam and accel grid currents flows to the screen grid, $I_s$. A current equal to the ion beam current is then emitted from the filament. The accel grid is biased negatively with respect to the filament to prevent electron backstreaming into the plasma source[5].

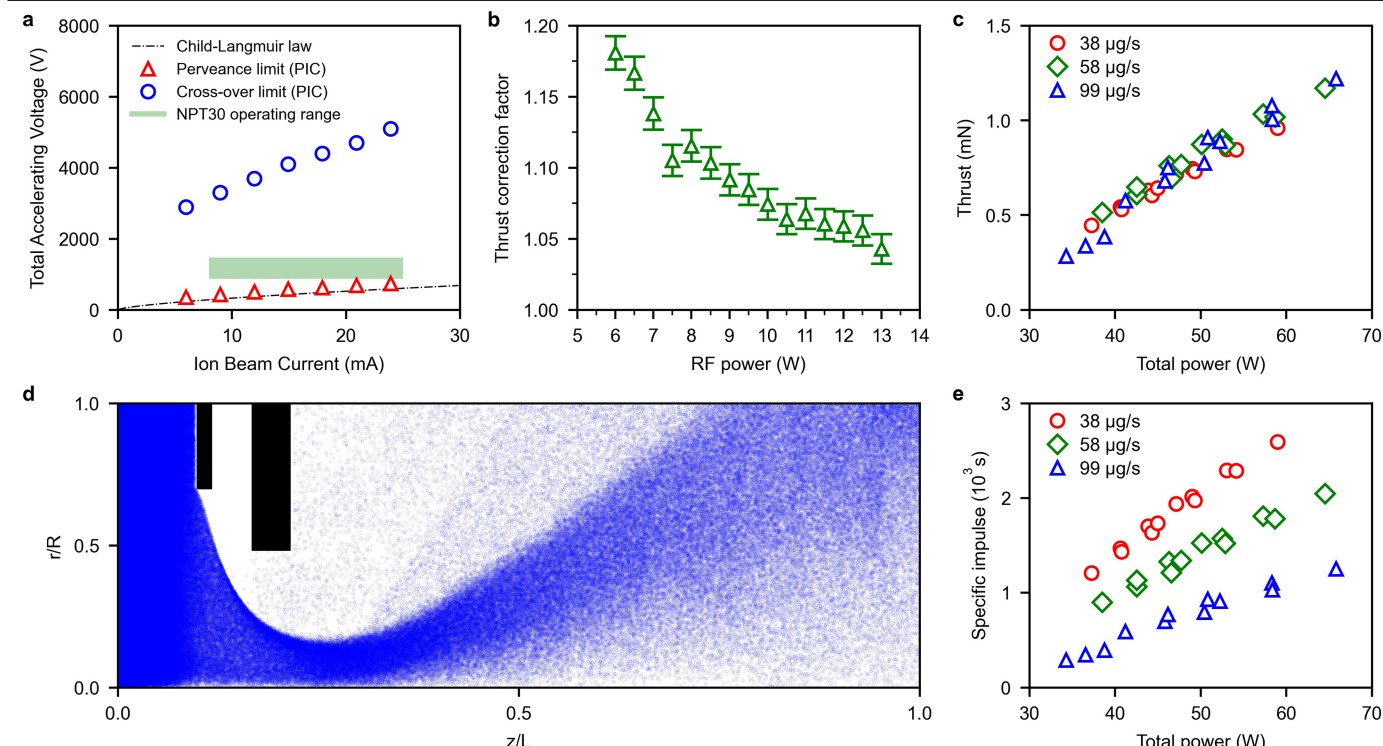

**Extended Data Fig. 3 | Ion optics simulations and propulsion system performance. a**, Perveance (space-charge) and cross-over limits of the propulsion system grid set obtained from particle-in-cell (PIC) simulations. The black dash-dot line shows the perveance limit from the Child-Langmuir law (see Methods), while the green shaded region denotes the operating range of the propulsion system. **b**, Correction factor as a function of RF power applied to the indirect thrust measurements to account for ion beam divergence and the presence of multiple ion species. **c**, Thrust of the propulsion system as a function of total system power and iodine input mass flow rate. **d**, PIC simulation of a single set of grid apertures (black shaded regions) showing the steady-state spatial ion distribution. The simulation is 2D in cylindrical coordinates and the domain has been normalized by the axial and radial simulation dimensions. **e**, Specific impulse of the propulsion system as a function of total system power and iodine mass flow rate. Error bars represent estimates of the measuring equipment precision and accuracy limitations.

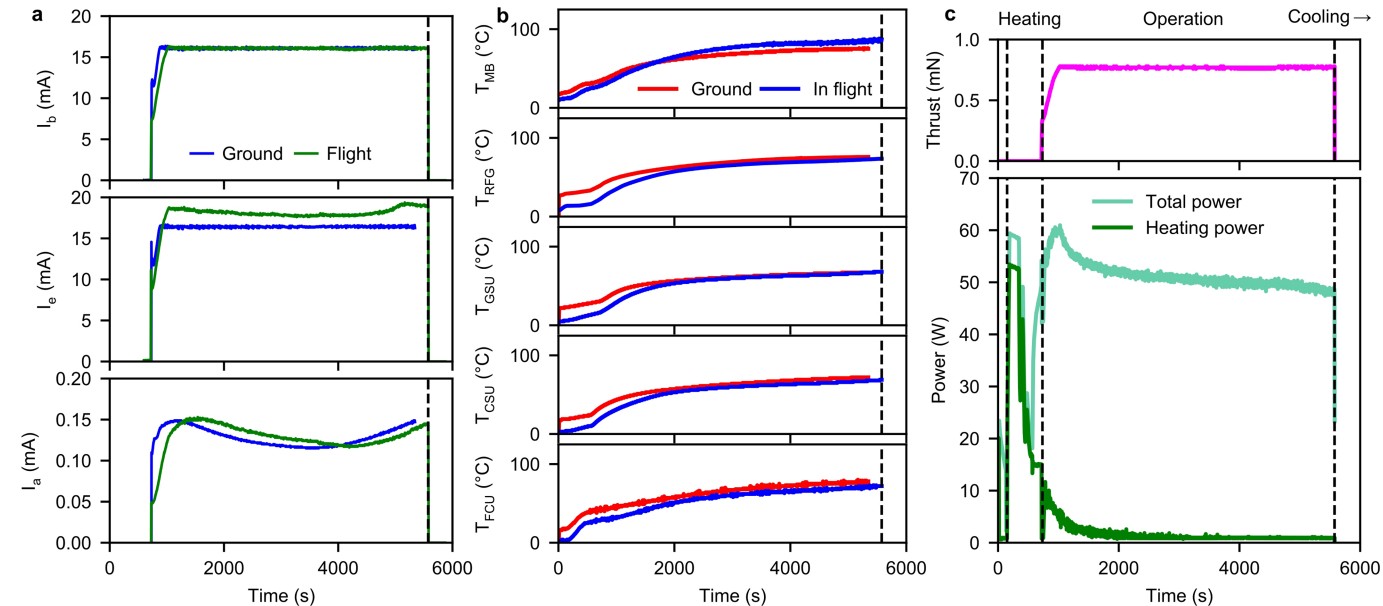

**Extended Data Fig. 4 | Ground-flight data comparison and propulsion system operation timeline. a**, Comparison between the ion beam current, $I_b$, electron neutralizer current, $I_e$, and current to the accel grid, $I_a$, during ground and in-flight operation for manoeuvre 1B. **b**, Measured electronic subsystem temperatures during propulsion system operation on the ground, and in space for manoeuvre 1B. The figure presents data for the main control unit, or Motherboard (MB), the Radio-Frequency Generator (RFG), the Grid Supply Unit (GSU), the Cathode Supply Unit (CSU), and the Flow Control Unit (FCU). **c**, In-orbit telemetry data of the thrust and power as a function of time for manoeuvre 1B indicating the propellant heating, propulsion system operation and propellant cooling stages.

**Extended Data Table 1 | Summary of propulsion system firing tests and resulting orbit changes**

| Firing slot | Date | Operation mode | Self-test and warm-up time (min) | Mean semi-major axis change (GMAT, m) | Mean semi-major axis change (GPS, m) |
|---|---|---|---|---|---|
| 1A | 29/12/2020 | FS | 9.2 | 299 (P) | 334 (P) |
| 1B | 02/01/2021 | N1 | 13.2 | 373 (P) | 368 (P) |
| 2A | 13/01/2021 | N1 | 11.2 | 272 (R) | 322 (R) |
| 2B | 14/01/2021 | N1 | 17.2 | 265 (P) | 411 (P) |
| 2C | 17/01/2021 | N1 | 11.5 | 427 (R) | 408 (R) |
| 2D | 21/01/2021 | N1 | 14.2 | 303 (R) | 310 (R) |
| 2E | 25/01/2021 | N1 | 12.1 | 200 (R) | 248 (R) |
| 2F | 28/01/2021 | N1 | 11.8 | 262 (R) | 265 (R) |
| 2G | 30/01/2021 | N1 | 11.5 | - | - |
| 2H | 09/02/2021 | N1 | 12.2 | 212 (R) | 232 (R) |
| 2I | 20/02/2021 | N1 | 11.6 | 232 (R) | 227 (R) |

See Methods for further details on the operational modes. The self-test and warm-up time is the total time needed before thrust generation begins. The labels (P) and (R) in the last two columns denote "prograde" and "retrograde", and indicate the direction of the thrust vector for each test. For test 2G, the satellite attitude control system changed the orientation of the thrust vector to be perpendicular to the satellite orbital plane. In this case, firing the thruster does not change the mean semi-major axis.