## [Peer Review File · Nature]

Manuscript Title: In-orbit demonstration of an iodine electric propulsion system

Reviewer Comments & Author Rebuttals

Reviewer Reports on the Initial Version:

Referee #1 (Remarks to the Author):

This is a well written technical paper on the performance and in-flight demonstration of the authors' iodine ion thruster. However, I don't feel that Nature is the appropriate journal for publication of this manuscript. Iodine ion thrusters have been around for many years and described in the literature. In addition to the authors previous publications going back several years, Busek Co. in the US demonstrated a very similar iodine thruster with comparable on-ground performance in 2017 (<http://spaceref.com/news/viewpr.html?pid=51284>), and also developed a similar flight system in 2019 (http://www.busek.com/index_htm_files/70010819F.pdf). The Busek iodine thruster system is scheduled to fly in Nov. 2021 on the Lunar IceCube Cubesat, shortly after this one. The first flight reported in this paper of an iodine ion thruster is then (as the title states) just a demonstration of a relatively well known technology. Nature normally publishes discoveries of major impact or interest. This paper is not a major scientific or engineering achievement, and is of interest only to the small cubesat community. I think it would a good paper in the Journal AIAA Spacecraft and Rockets, but doesn't rise to stature of Nature. I therefore have to unfortunately recommend rejection.

Some specific comments for the authors to consider before resubmission.

- The thrust and total power of the system in flight should be given in the abstract (in-orbit it seems like 0.8 mN at 55 W are mentioned most).
- Most journals don't like references in the abstract.
- The first four references are an odd choice since they are supposed to support the statement, "Propulsion is a critical subsystem of many spacecraft needed for orbit transfer maneuvers or compensation of aerodynamic or gravitational perturbations." These references don't do that.
- I disagree with the notion that xenon and krypton are expensive. SpaceX has now launched over 1500 Starlink satellites with krypton-fueled Hall thrusters, so it can't be that expensive. Xenon for larger satellites is still a tiny fraction ($\ll 1\%$) of the total mission cost. If the authors are going to make this statement, they need to show that an iodine-fueled spacecraft is less expensive than a xenon or krypton-fueled one, especially after accounting for the impacts of iodine on the rest of the flight system. Arguments about tank size are valid, but cost reduction needs to be justified.
- Seems like the introduction is missing any mention of Deep Space 1, BepiColombo, and the Starlink satellites as electric propulsion missions.
- Page 4 says, "A cathode filament downstream of the grids thermionically emits electrons to charge neutralize the ion beam." This doesn't seem like a long-term solution because of sputter erosion lifetime concerns which drove the invention of plasma bridge neutralizer hollow cathodes back in the 1970's. The total impulse given seems to imply about 1500 h life. Is this limited by the neutralizer filament life or some other factor?
- Page 6: Heating a propellant tank to 80-100C would be non-trivial for a large amount of propellant. There's also a heating cycle to get the iodine out of the reservoir. In a typical 90 min hr burn, how much propellant is lost before and after the electric thrust is operated? It seems like a modified mass utilization efficiency is needed to handle this wasted propellant before and after the ion beam is turned on. Also, a timeline of the operation showing the heating, ignition, operation and cooling phase durations in the thruster operation should be provided.
- Page 6: "...the iodine is embedded into a rigid porous matrix." What is the resulting tankage fraction?
- There is a lack of information in the manuscript on the total power required to run this system, which is a critical concern for cubesat and small-sat applications. There's mass utilization versus

total power in Fig. 2, but not thrust, Isp or efficiency versus total power. For the ground data, Fig. 3 should have thrust versus total system power (not just the rf power), and perhaps total system efficiency versus total input power.

- Figure 2 shows beam current and beam composition at rf power levels below 14 W, and yet all the in-flight performance discussion is at around 50-60 W. These graphs should extend to the in-flight power levels.
- Figure 4d is a comparison of in-flight and ground data for 6 parameters and situations. The graph is too small to be able to see any comparison, and needs to be pulled out of Fig. 4 and made into a separate full size figure by itself so we can see the data and comparisons.

Referee #2 (Remarks to the Author):

The paper "First in-orbit demonstration of an iodine electric propulsion system," by Rafalskyi et al., describes a first flight of an iodine-fed electric propulsion system. The paper discusses the thruster design and diagnostics used to quantify the thruster's performance on the ground and in flight, with comparisons between the two to show the efficacy of the propulsion system. The comparisons of flight propulsion data, ground test data, and in-flight tracking telemetry is fantastic and definitely serves to illustrate thruster performance. First flights of propulsion systems are always novel and deserving of publication. This is an excellent paper, it was a pleasure to be asked to review it, and I can enthusiastically endorse its publication. There are a few minor comments offered below, in the hopes that they will help the authors.

1) The abstract material is good, but typically abstracts should be written in past tense passive voice, with I, we, our avoided. The only minor quibble on the substance of the abstract is in the statement (lines 26-27) that "results demonstrate for the first time that iodine is a viable propellant for electric propulsion systems." The results demonstrated this 'in flight' for the first time, but previous work by several others, while not leading to a flight as of this writing, did demonstrate the viability of iodine as an electric propulsion propellant.

2) Page 3, line 48, is the reference on 'Electric propulsion' supposed to be 5,15 or 5-15?

3) Page 3, line 53, no comma needed between 'gas' and 'and'.

4) Page 3, lines 63, 64, no commas needed between 'accelerators' and 'requires' and 'threshold' and 'and'. (There are many examples of unneeded commas throughout the manuscript. This reviewer will not call out every instance in this review, but the document should be reviewed by the authors and editorial staff to remove these unnecessary punctuation marks prior to publication.)

5) Page 4, lines 71-72, the statement that xenon "is incompatible with the "new space" paradigm." is interesting. Is this because of all the things previously stated (that its high acquisition cost, the need for specialized equipment and trained personnel (also implying high cost for processing), and the high storage pressure (which implies that it may not be welcome as a secondary payload), or are there other additional reasons why this is the case? If the former, recommend recasting the statement ", making it incompatible with the "new space" paradigm." If the latter, recommend spelling out exactly why it is incompatible.

5) Page 4, line 76, 'it has both a similar ionization'... addition of 'both' to let the reader know that the similarity applies to the ionization threshold and the relative atomic mass.

6) Page 7, lines 140-141, is it really true to say "higher iodine dissociation occurs at higher powers due to an increased plasma density." The higher power likely results in a higher plasma density, but doesn't the higher power per unit mass also translate to greater dissociation (in other words,

both the increased plasma density and the higher iodine dissociation are due to the increased input power per unit mass of propellant).

7) Page 11, around line 221, the duration of the firing (with the time for iodine heating) is given. What is the duration of actual thrusting during these firings? Also, the caption for Fig. 4 is separated from the actual figure.

8) Page 18, lines 388-390, it might be good to specifically call these out as 'charge exchange' collisions.

9) Page 25, line 533, the rectangular pulse width is very short. Does the current collected at the probe have time to reach a steady-state value for this short a pulse? If so, how long does it take to reach steady state?

10) Page 26, direct thrust measurement section. I'm curious about the thrust stand implementation described. If you have a pendulum-based thrust stand, typically the measurement is on the position with a known (or measured) force applied to calibrate. The description given seems to imply that there is a pendulum-based thrust stand with a load cell on the end measuring the direct force applied by the thruster. That would be slightly different from the typical method and I'm hoping the authors can provide a few more details to clear up the confusion.

11) Page 6, related to lines 116-118. The use of iodine as a control valve blocking the flow of iodine is innovative, and I would've thought that having solid iodine exposed to vacuum would've made this problematic. It was hoped that a few additional words would've been included on how this worked, including how much propellant was lost as the iodine exposed to vacuum sublimed and/or as the feed system was heated or cooled prior to and after firing.

Referee #3 (Remarks to the Author):

This report is about the manuscript "First in-orbit demonstration of an iodine electric propulsion system" by Dmytro Rafalskyi and coworkers. The text is coherently written and shows for the first time that it is possible to perform manoeuvres in space with an iodine-powered ion engine, which may become a game-changer for some space missions. It is therefore very likely to have a high impact on the field of electric space propulsion. This is all the more significant due to NASA's highly publicised iodine engine demonstration mission (iSAT-1 (Iodine Hall, iodine Satellite), which was cancelled due to difficulties in operating the engine. There are no methodological flaws or dubious handling of data and the quality of data is high. In my opinion, the manuscript meets the general requirements for publication in Nature. However, I would like to share the following points with the authors before publication, which I believe would further improve the quality of the manuscript.

The EP community has certainly relied mainly on xenon as a propellant for the last few years. Given the fact that the Starlink mega-constellation has already put over 1600 satellites into orbit in a very short time using krypton as propellant, I dare say that krypton is the real propellant of choice - or at least will be in the next 1-2 years. Although I do not know the exact amount of krypton on board of such a satellite, it will be in the order of a few kilograms, which means that krypton could also become a rare good in the near future. This aspect should be emphasised more, as it shows all the more the importance of cheap propellant alternatives that are available in sufficient quantities. What is no longer true is the claim that xenon is massively used in car headlights. LED lamps took over here some time ago.

Re line 75: The comparison of expensive xenon and cheap iodine seems to me to be a little too brief. In my experience, the price of iodine depends very much on the purity used. High-purity iodine plays in the same price range as high-purity xenon, is sometimes even more expensive and

may not be available in large quantities per se. I would like to see this aspect discussed in more depth.

On line 76: In my opinion, this is where it gets a bit fuzzy. Both the iodine atom and the iodine molecule have a lower ionisation threshold (10.5 and 9.4 eV). Whether this should be called similar in size to xenon (12.1 eV) may be a matter of taste. For me, it is a clear difference of approx. 15 and 28 %, respectively. Since iodine is fed into the plasma as a molecule, its mass is also twice as high as that of xenon. Here I would suggest a little more care.

Re line 93: The engine is apparently operated with a thermionic electron emitter. These are usually not very durable. Furthermore, they may only neutralize the beam of a small ion thruster emitting of few ten mA of ions. For large thrusters emitting several A of ions, this approach may not be efficient enough. Additionally, what about the service life of the thermionic emitter? Operating a thruster for a short time (even if it happens in space) is certainly not the greatest challenge. It is the longevity of these engines that is the challenging part. The authors should perhaps discuss this point in a little more depth. Especially since the technology of inductively coupled rf plasmas used by the authors also offer a solution here, as was recently shown by Dietz et al. (<https://doi.org/10.1051/epjap/2020190213>). In addition, what kind of operating times are being aimed for or are possible?

Line 118: It would be interesting to know how this "innovative" regulation works, at least according to which physical aspects. It sounds more like an advertisement for a product without knowing how it works.

Line 120: This sentence should be clarified. The term "fragmentation" may easily be confused with a dissociation of the molecule.

Line 122: Here reference is made to a porous matrix. Is there a bit more information possible? It sounds like an advertising brochure again.

Line 134: I would like to emphasize that the more direct process $I_2 + e^- \rightarrow I^+ + I + 2e^-$ is also possible and perhaps even more likely at rather large electron temperatures (See publication by Pascaline Grondein et al. (Ref. 29)).

Line 150: How has the mass flow been calibrated? From previous publications, e.g. Ref. 30; one would expect a similar beam current at the same mass flow and rf power. The same is true for the simulation results given by Grondein et al (Ref. 29).

Figure 2: Which RF power is given here? The power provided by the RFG or the input power of the RFG? While the former is more useful in terms of plasma physics, the latter is more important for the actual application. Some uncertainties seem to be very large, for e.g a range from about 5 % to 15 % of the mass efficiency (a factor of 3) at about 34 W and 99 $\mu\text{g/s}$. What are the reasons for this partially very large uncertainties?

Figure 3: a) is the IEDF, not the IVDF. The IVDF should contain multiple peaks due to several m/q ratios.

Line 358: what exactly means global mass? And what is a global continuity?

Line 369: Is a mean mass used to calculate the Bohm velocity or is the velocity calculated for each ion species? The used approach may influence the results quite significantly.

Line 407: Which ion mass was assumed for calculating the perveance? Since the perveance is proportional to the inverse square root of the ions mass, this should at least be mentioned.

Line 416: This factor should be introduced directly with the Eq. in line 176. Perhaps it should be mentioned that the pre-factors simply represent the root of the mass-charge ratio to the iodine atom.

Line 470: The actual required heating power may be much larger due to radiative losses compared to the power required for sublimation. This should be discussed at least by an example.

Lines 502-506: What is estimated pumping speed for atomic and molecular iodine? The correction factor for iodine may suffer large uncertainties, since the composition (I and I₂) of the neutral gas close to the pressure gauge may not be known well. How was this issue solved?

Line 513: Have negative iodine ions been detected in the beam? Negative iodine ions may not be able to overcome the sheath potential and should mainly remain in the discharge until recombination. Or is there a high amount of electron-capture processes between neutral iodine and electrons?

Lines 527-537: What is the distance between thruster (grid) and molybdenum disk? Normally, a TOF measurement requires a fast stop signal. A simple current probe seems to me being a slow device with high time constants. Is there any additional/supporting technique used to increase quality of the TOF spectra? What is the rise and fall time of the high voltage pulse?

Lines 552-554: It is claimed that a retarding field energy analyzer measures the velocity function and not the energy distribution function. That is not correct. Publications on RFEAs consistently claim that the IEDF is measured (see for instance Gahan et al. "Retarding field analyzer for ion energy distribution measurements at a radio-frequency biased electrode", Review of Scientific Instruments 79, 033502 (2008); <https://doi.org/10.1063/1.2890100>). And anyway, what is a velocity supposed to be in energy units? A velocity (or momentum) filter normally requires a magnetic field. Furthermore, to be more concrete: the RFEA determines the distribution function of the accelerating potential.

One last point on axis labelling for graphs. I am not aware of any further specifications from Nature here and in my opinion it is not my task either, but for the sake of completeness I want to point out that units in square brackets is the only notation that is rejected in all valid standards (e.g. in ISO 31-0). I would recommend a standard-compliant variant here.

Author Rebuttals to Initial Comments:

Referee #1

General comments

This is a well written technical paper on the performance and in-flight demonstration of the authors' iodine ion thruster. However, I don't feel that Nature is the appropriate journal for publication of this manuscript. Iodine ion thrusters have been around for many years and described in the literature. In addition to the authors previous publications going back several years, Busek Co. in the US demonstrated a very similar iodine thruster with comparable on-ground performance in 2017 (<http://spaceref.com/news/viewpr.html?pid=51284>), and also developed a similar flight

system in 2019 (http://www.busek.com/index_htm_files/70010819F.pdf). The Busek iodine thruster system is scheduled to fly in Nov. 2021 on the Lunar IceCube Cubesat, shortly after this one. The first flight reported in this paper of an iodine ion thruster is then (as the title states) just a demonstration of a relatively well known technology. Nature normally publishes discoveries of major impact or interest. This paper is not a major scientific or engineering achievement, and is of interest only to the small cubesat community. I think it would a good paper in the Journal AIAA Spacecraft and Rockets, but doesn't rise to stature of Nature. I therefore have to unfortunately recommend rejection.

Some specific comments for the authors to consider before resubmission.

General response

We thank the referee for their review and their useful comments to help us improve our work. While it is true that some other organisations (including Busek which we originally cited) have developed iodine propulsion systems, none have yet been tested in space. The well-known NASA Iodine Satellite (iSat) mission for example was scheduled to be launched in 2018 but was cancelled due to problems in operating the thruster with iodine. We believe therefore that successful operation in space should not be viewed as a given. We also believe that our work has a much broader interest outside of the CubeSat community since it both highlights the importance of alternative propellants, and directly demonstrates the feasibility of iodine to replace xenon. We hope that our manuscript catalyses further research within the space community and accelerates the adoption of new propellants.

Comment 1

The thrust and total power of the system in flight should be given in the abstract (in-orbit it seems like 0.8 mN at 55 W are mentioned most).

Response 1

Nature guidelines usually specify not to include numbers within the abstract. Since these numbers are not critical to the abstract, and since our original abstract was slightly too long anyway, we have chosen not to add them. We have however added these numbers in the revised manuscript when first introducing the propulsion system.

Comment 2

Most journals don't like references in the abstract.

Response 2

Here we have followed Nature guidelines which ask for a fully referenced abstract. This has been confirmed with the Editor.

Comment 3

The first four references are an odd choice since they are supposed to support the statement, "Propulsion is a critical subsystem of many spacecraft needed for orbit transfer maneuvers or compensation of aerodynamic or gravitational perturbations." These references don't do that.

Response 3

Reference 1 was chosen as it discusses the general use of propulsion for all in-space mobility requirements, while references 3-4 were chosen because propulsion is a critical subsystem needed for the success of each of these missions. In retrospect, we agree that our placement of the citation at the end of the sentence is perhaps confusing. We have modified this sentence in our revised manuscript to remove this confusion by shortening it since the abstract length is slightly too long anyway.

Comment 4

I disagree with the notion the xenon and krypton are expensive. SpaceX has now launched over 1500 Starlink satellites with krypton-fueled Hall thrusters, so it can't be that expensive. Xenon for larger satellites is still a tiny fraction (<<1%) of the total mission cost. If the authors are going to make this statement, they need to show that an iodine-fueled spacecraft is less expensive than a xenon or krypton-fueled one, especially after accounting for the impacts of iodine on the rest of the flight system. Arguments about tank size are valid, but cost reduction needs to be justified

Response 4

SpaceX is targeting a total cost per Starlink satellite of less than \$500k-\$1M. Their choice to shift from xenon Hall thrusters to krypton is because of the large cost of xenon, the large amount of xenon

needed (estimated at 12 metric tons per year, which is almost 20% of the global xenon production output), and the likely increase in xenon prices with time due to demand from both other propulsion organisations, and the increasing demand from the semiconductor industry. Since iodine is not yet a mature enough propellant, SpaceX was more or less forced to use krypton as an alternative. It is difficult to find accurate numbers on the propellant mass used on Starlink satellites, but it is estimated to be between about 10-30 kg, which comes out to a propellant cost alone of between \$35k-\$105k per satellite if xenon were used. By contrast, with krypton the estimated cost is \$2.5k-7.5k.

Krypton however has a higher ionization threshold than xenon, and a lower atomic mass which reduces its thrust-to-power ratio. Both of these factors mean about 25% or higher power is needed for a given thrust. Krypton also has a storage density about 3 times lower than xenon at the same temperature, which increases the size and mass of the propellant tank.

In our revised manuscript, based on the referee's comment, and a related comment by Referee #3, we have added a new section to Methods that includes an iodine vs xenon cost comparison, and identifying krypton as an alternative propellant in the short-term.

Comment 5

Seems like the introduction is missing any mention of Deep Space 1, BepiColombo, and the Starlink satellites as electric propulsion missions.

Response 5

Due to manuscript size limitations, we were not able to mention all major electric propulsion missions. References to Deep Space 1 and BepiColombo are however already included in Refs. [3] and [4]. While Starlink is a very prominent recent application of electric propulsion, there are no scientific references available from SpaceX itself. However, in the revised manuscript we now discuss the Starlink satellites in a new section in Methods and cite a reference that has analysed the propellant costs of the Starlink satellites.

Comment 6

Page 4 says, "A cathode filament downstream of the grids thermionically emits electrons to charge neutralize the ion beam." This doesn't seem like a long-term solution because of sputter erosion lifetime concerns which drove the invention of plasma bridge neutralizer hollow cathodes back in the 1970's. The total impulse given seems to imply about 1500 h life. Is this limited by the neutralizer filament life or some other factor?

Response 6

We agree with the referee. Since the propulsion system in our manuscript is low power with a low ion beam current, a thermionic neutralizer was chosen because it allows significant miniaturization of the system, and reduces the additional mass and propellant flow needed for conventional hollow cathode plasma bridge neutralizers.

The total burn time of 1500 hours is limited by the amount of propellant that can be accommodated in a 1-unit CubeSat form factor. A second version of the propulsion system has been developed with a 1.5-unit CubeSat form factor that extends the propellant tank and increases the total impulse to 9500 Ns, or just over 2500 hours of total burn time. By comparison, the life of the neutralizer is limited to about 3600 hours. We have modified the revised manuscript and added a new section to Methods to more clearly state the reason for using a thermionic filament neutralizer.

Comment 7

Page 6: Heating a propellant tank to 80-100C would be non-trivial for a large amount of propellant. There's also a heating cycle to get the iodine out of the reservoir. In a typical 90 min hr burn, how much propellant is lost before and after the electric thrust is operated? It seems like a modified mass utilization efficiency is needed to handle this wasted propellant before and after the ion beam is turned on. Also, a timeline of the operation showing the heating, ignition, operation and cooling phase durations in the thruster operation should be provided.

Response 7

We agree with the referee that depending on the size of the propellant tank heating may be difficult. However, this depends significantly on the exact design of the system which is no longer constrained by high-pressure storage requirements as with some other propellants. Additionally, for larger thrusters more waste heat is available for heating.

The mass flow rate depends on the vapor pressure which is a strong function of the iodine temperature. During most of the heating cycle the mass loss is quite low and the flow rate only increases significantly as the nominal operating temperatures between 80-100 °C are reached just before thrust is initiated. Based on ground testing, the estimated propellant mass loss is around 12 mg during each on/off cycle. This mass loss has been added to the revised manuscript.

For the in-orbit demonstration in our manuscript, the duration of the burn was limited by the spacecraft power generation capability. If more power was available, the propulsion system could be operated continuously. A modified mass utilization efficiency or specific impulse could be used to handle this mass loss, and we recently discussed this in a more general context in Ref. [45] (original manuscript, [48] in the revised manuscript), since the mass loss during startup is not just a consideration in our system, but can be important for many other propulsion systems as well.

A figure showing the timeline of propulsion system operation has been added to the revised manuscript. Due to size limits of the main manuscript text, this figure has been added to the Extended Data Figures section.

Comment 8

Page 6: "...the iodine is embedded into a rigid porous matrix." What is the resulting tankage fraction?

Response 8

The tankage fraction (defined as the total tank mass, including the porous matrix, divided by the propellant mass) is approximately 54%. This information has now been added to the revised manuscript.

Comment 9

There is a lack of information in the manuscript on the total power required to run this system, which is a critical concern for cubesat and small-sat applications. There's mass utilization versus total power in Fig. 2, but not thrust, Isp or efficiency versus total power. For the ground data, Fig. 3 should have thrust versus total system power (not just the rf power), and perhaps total system efficiency versus total input power.

Response 9

Thrust and specific impulse versus total power lower and upper limits is indicated in Figure 3 (d). We have now modified this figure to add additional total power labels. We have also added additional figures showing the thrust and specific impulse versus total power. Due to size limits of the main manuscript text, these figures have been added to the Extended Data Figures section.

Comment 10

Figure 2 shows beam current and beam composition at rf power levels below 14 W, and yet all the in-flight performance discussion is at around 50-60 W. These graphs should extend to the in-flight power levels.

Response 10

We were not precise enough with our explanation of power. The RF power in Figure 2 corresponds to the RF power supplied by the RF generator, whereas the total power between 50-60 W includes all power consumed by the propulsion system, including the RF power, acceleration power supplied to the grids, the neutralizer power, the power to run the onboard electronics, all heating power required to heat the iodine propellant, and all PPU losses. Thus, although the power is around 50-60 W for the in-flight performance, the fraction of this power that corresponds to the RF power is below 14 W, as displayed in Figure 2. We have modified the text in the revised manuscript to more clearly define the powers used.

Comment 11

Figure 4d is a comparison of inflight and ground data for 6 parameters and situations. The graph is too small to be able to see any comparison, and needs to be pulled out of Fig. 4 and made into a separate full size figure by itself so we can see the data and comparisons.

Response 11

Because of size limits of the main manuscript text, we have added an extra separate figure of Figure 4 (d) to the Extended Data Figures section of the manuscript. Since we feel Figure 4 (d) is still relevant to the results in Figure 4, we have decided to leave this figure unchanged.

Referee #2

General comments

The paper "First in-orbit demonstration of an iodine electric propulsion system," by Rafalskyi et al., describes a first flight of an iodine-fed electric propulsion system. The paper discusses the thruster design and diagnostics used to quantify the thruster's performance on the ground and in flight, with comparisons between the two to show the efficacy of the propulsion system. The comparisons of flight propulsion data, ground test data, and in-flight tracking telemetry is fantastic and definitely serves to illustrate thruster performance. First flights of propulsion systems are always novel and deserving of publication. This is an excellent paper, it was a pleasure to be asked to review it, and I can enthusiastically endorse its publication. There are a few minor comments offered below, in the hopes that they will help the authors.

General response

We thank the referee for their positive review and appreciate their useful comments to help us improve the quality of our manuscript.

Comment 1

The abstract material is good, but typically abstracts should be written in past tense passive voice, with I, we, our avoided. The only minor quibble on the substance of the abstract is in the statement (lines 26-27) that "results demonstrate for the first time that iodine is a viable propellant for electric propulsion systems." The results demonstrated this 'in flight' for the first time, but previous work by several others, while not leading to a flight as of this writing, did demonstrate the viability of iodine as an electric propulsion propellant.

Response 1

We have tried to follow Nature guidelines which make use of a first-person perspective in the abstract. We have checked this point with the Editor, and so have chosen to leave the abstract with the words "we" and "our".

Since the abstract is slightly too long, we have chosen to remove the sentence in lines 26-27 highlighted by the referee.

Comment 2

Page 3, line 48, is the reference on 'Electric propulsion' supposed to be 5,15 or 5-15?

Response 2

Although references 5-15 are all related to electric propulsion, our intent was for the reference to mean “5,15” since references 5 and 15 are fundamental textbooks introducing the field of electric propulsion.

Comment 3

Page 3, line 53, no comma needed between 'gas' and 'and'.

Response 3

The comma has been removed in our revised manuscript.

Comment 4

Page 3, lines 63, 64, no commas needed between 'accelerators' and 'requires' and 'threshold' and 'and'. (There are many examples of unneeded commas throughout the manuscript. This reviewer will not call out every instance in this review, but the document should be reviewed by the authors and editorial staff to remove these unnecessary punctuation marks prior to publication.)

Response 4

We have removed these commas and also performed a more careful proofread of the revised manuscript to remove other unnecessary commas.

Comment 5

Page 4, lines 71-72, the statement that xenon "is incompatible with the "new space" paradigm." is interesting. Is this because of all the things previously stated (that its high acquisition cost, the need for specialized equipment and trained personnel (also implying high cost for processing), and the high storage pressure (which implies that it may not be welcome as a secondary payload), or are there other additional reasons why this is the case? If the former, recommend recasting the statement ", making it incompatible with the "new space" paradigm." If the latter, recommend spelling out exactly why it is incompatible.

Response 5

Our intent was the former, so we have reformulated the sentence as suggested by the referee.

Comment 6

Page 4, line 76, 'it has both a similar ionization'... addition of 'both' to let the reader know that the similarity applies to the ionization threshold and the relative atomic mass.

Response 6

Based on other comments from Referee #3 we have reformulated this sentence to more carefully compare iodine and xenon.

Comment 7

Page 7, lines 140-141, is it really true to say "higher iodine dissociation occurs at higher powers due to an increased plasma density." The higher power likely results in a higher plasma density, but doesn't the higher power per unit mass also translate to greater dissociation (in other words, both the increased plasma density and the higher iodine dissociation are due to the increased input power per unit mass of propellant).

Response 7

Since the pressure in the source tube during operation is low, plasma-gas heating is expected to also be relatively low (a few 100's °C; see for example P. Chabert et al, *Physics of Plasmas* **19**, 073512

(2012)] so that direct thermal dissociation of diatomic iodine is not significant [R.T. Holbrook and J.A. Kunc, *Physics of Plasmas* **1**, 1075 (1994)]. In this case, dissociation only occurs due to electron impact collisions with diatomic iodine molecules, and so the dissociation rate is proportional to the electron/plasma density.

Comment 8

Page 11, around line 221, the duration of the firing (with the time for iodine heating) is given. What is the duration of actual thrusting during these firings? Also, the caption for Fig. 4 is separated from the actual figure.

Response 8

The total propulsion system startup up time is between about 10-20 min, so that the actual firing time is 60-80 min. This has now been clarified in the text, and the caption for Fig. 4 has been moved closer to the figure.

Comment 9

Page 18, lines 388-390, it might be good to specifically call these out at 'charge exchange' collisions.

Response 9

Although we agree with the referee that charge-exchange collisions are expected to be the dominate collisional process, we cannot rule out yet the contribution from downstream ionization in the plume. That is to say, electrons from the neutralizer can generate a secondary plasma in the plume from any un-ionized neutral gas, and the low-energy ions in this secondary plasma could then also strike the grid. There is also the possibility of negative ion formation due to the high electron attachment cross-section at low electron energies [D.C. Frost and C.A. McDowell, *Canadian Journal of Chemistry* **38**, 407 (1960)]. Thus, the plume physics may be relatively complicated and the exact dominant processes not yet known. Nonetheless, we have modified lines 388-390 to specifically mention some of the above possibilities.

Comment 10

Page 25, line 533, the rectangular pulse width is very short. Does the current collected at the probe have time to reach a steady-state value for this short a pulse? If so, how long does it take to reach steady state?

Response 10

For the time-of-flight measurements, our objective is to estimate the transit time of different ions species extracted from the propulsion system to the probe. In this case, steady-state operation is to be avoided since it would then not be possible to distinguish ions with different q/m ratios. That is to say, steady-state measurements would result in only a single nominal current value from which it is not possible to determine the different ion species contributions. The only way to distinguish different q/m ratios is by operating the probe in a transient manner since in this way ions with a different q/m ratio will have a different velocity, and hence the transit time to the probe changes.

Comment 11

Page 26, direct thrust measurement section. I'm curious about the thrust stand implementation described. If you have a pendulum-based thrust stand, typically the measurement is on the position with a known (or measured) force applied to calibrate. The description given seems to imply that there is a pendulum-based thrust stand with a load cell on the end measuring the direct force applied by the thruster. That would be slightly different from the typical method and I'm hoping the authors can provide a few more details to clear up the confusion.

Response 11

Yes, the referee is correct. Our thrust stand uses a load cell at one end to measure the thrust force. The propulsion system is attached to the bottom of the pendulum and the thrust force at the end of the pendulum is then transferred to the force sensor. The sensor and thrust vector location are shifted which changes the respective pendulum lever arms and allows the measured force on the sensor to be magnified compared to the thrust force itself. The balance was calibrated with a set of known test masses on a horizontal arm producing a known moment. We have modified the revised manuscript to add some of these details.

Comment 12

Page 6, related to lines 116-118. The use of iodine as a control valve blocking the flow of iodine is innovative, and I would've thought that having solid iodine exposed to vacuum would've made this problematic. It was hoped that a few additional words would've been included on how this worked, including how much propellant was lost as the iodine exposed to vacuum sublimed and/or as the feed system was heated or cooled prior to and after firing.

Response 12

The flow of iodine is blocked with an insert between the tank and source tube that contains a sub-millimetre hole. This configuration has a low thermal conductive coupling with the propellant tank. When the propulsion system is not operating, the insert rapidly cools and causes the saturated vapour pressure at the walls to drop sharply, which in turn causes iodine to deposit blocking the hole and preventing iodine vapour in the tank from exiting. In addition, the effective flow conductance is significantly reduced because of the design of the orifice, gas distribution head, source tube, and acceleration grids themselves, so that the iodine leakage rate is also low. Ground-based experiments have been performed with the propulsion system off and stored under vacuum for over 2 weeks. Measurements of the mass loss were at the resolution limit of our mass balance (0.1 g), and indicate an upper limit total mass loss of less than 0.1 g, corresponding to an upper limit leakage rate of 0.08 $\mu\text{g/s}$. We have modified the text in the revised manuscript to provide more information on the flow control and leakage rate.

Referee #3

General comments

This report is about the manuscript "First in-orbit demonstration of an iodine electric propulsion system" by Dmytro Rafalskyi and coworkers. The text is coherently written and shows for the first time that it is possible to perform manoeuvres in space with an iodine-powered ion engine, which may become a game-changer for some space missions. It is therefore very likely to have a high impact on the field of electric space propulsion. This is all the more significant due to NASA's highly publicised iodine engine demonstration mission (iSAT-1 (Iodine Hall, iodine Satellite), which was cancelled due to difficulties in operating the engine. There are no methodological flaws or dubious handling of data and the quality of data is high. In my opinion, the manuscript meets the general requirements for publication in Nature. However, I would like to share the following points with the authors before publication, which I believe would further improve the quality of the manuscript.

General response

We would like to thank the referee for their positive review, and we appreciate their comments to help us improve our manuscript.

Comment 1

The EP community has certainly relied mainly on xenon as a propellant for the last few years. Given the fact that the Starlink mega-constellation has already put over 1600 satellites into orbit in a very short time using krypton as propellant, I dare say that krypton is the real propellant of choice - or at least will be in the next 1-2 years. Although I do not know the exact amount of krypton on board of such a satellite, it will be in the order of a few kilograms, which means that krypton could also become a rare good in the near future. This aspect should be emphasised more, as it shows all the more the importance of cheap propellant alternatives that are available in sufficient quantities. What is no longer true is the claim that xenon is massively used in car headlights. LED lamps took over here some time ago.

Response 1

The referee raises a number of good points. We have modified the revised manuscript to discuss krypton in more detail, but due to size limits of the main text this has been done by adding a new section to Methods.

Although it is true that LED lamps are taking over, the use of xenon in imaging and lighting applications still accounts for just over 20% of the market. We have rephrased the sentence and removed explicit mention of car headlights. We have however added a new application and reference to semiconductor etching, which is estimated to be one of the fastest growing market segments.

Comment 2

Re line 75: The comparison of expensive xenon and cheap iodine seems to me to be a little too brief. In my experience, the price of iodine depends very much on the purity used. High-purity iodine plays in the same price range as high-purity xenon, is sometimes even more expensive and may not be available in large quantities per se. I would like to see this aspect discussed in more depth.

Response 2

In the development of our propulsion system we used both xenon and iodine propellants and so are able to provide direct cost numbers. Due to size limits of the main text, we have added a new short section to Methods comparing these costs.

High purity iodine is not needed in our propulsion system, and both ground testing and in-orbit operation was conducted with 99.5% pure iodine.

Comment 3

On line 76: In my opinion, this is where it gets a bit fuzzy. Both the iodine atom and the iodine molecule have a lower ionisation threshold (10.5 and 9.4 eV). Whether this should be called similar in size to xenon (12.1 eV) may be a matter of taste. For me, it is a clear difference of approx. 15 and 28 %, respectively. Since iodine is fed into the plasma as a molecule, its mass is also twice as high as that of xenon. Here I would suggest a little more care.

Response 3

After rereading this line, we see the referee's point and have modified the text in the revised manuscript to more carefully compare iodine and xenon.

Comment 4

Re line 93: The engine is apparently operated with a thermionic electron emitter. These are usually not very durable. Furthermore, they may only neutralize the beam of a small ion thruster emitting of few ten mA of ions. For large thrusters emitting several A of ions, this approach may not be efficient enough. Additionally, what about the service life of the thermionic emitter? Operating a thruster for a short time (even if it happens in space) is certainly not the greatest challenge. It is the longevity of these engines that is the challenging part. The authors should perhaps discuss this point in a little more depth. Especially since the technology of inductively coupled rf plasmas used by the authors also offer a solution here, as was recently shown by Dietz et al. (<https://doi.org/10.1051/epjap/2020190213>). In addition, what kind of operating times are being aimed for or are possible?

Response 4

We agree with the referee. Since the propulsion system in our manuscript is low power with a low ion beam current, a thermionic neutralizer was chosen because it allows significant miniaturization of the system, and reduces the additional mass and propellant flow needed for conventional hollow cathode plasma bridge neutralizers.

For the current propulsion system, the maximum impulse that can be delivered is 5500 Ns, which corresponds to a total burn time of just over 1500 hours. The propulsion system has two thermionic neutralizers for a total estimated lifetime of 3600 hours. Information on the total burn time and lifetime is now mentioned within the paper.

We appreciate the paper highlighted by the referee, and we had already commenced work in a similar direction for neutralizing ion beams from larger propulsion systems.

Comment 5

Line 118: It would be interesting to know how this "innovative" regulation works, at least according to which physical aspects. It sounds more like an advertisement for a product without knowing how it works.

Response 5

When the propulsion system is not active, the flow path is no longer heated and iodine vapour begins to deposit and solidify. This deposition blocks the small orifice between the propellant tank and the plasma discharge chamber stopping the flow of any remaining iodine vapour within the propellant tank. Thus, the phase change of iodine itself is used as a "control valve" instead of a solenoid valve. A new section has been added to Methods providing further details.

Comment 6

Line 120: This sentence should be clarified. The term "fragmentation" may easily be confused with a dissociation of the molecule.

Response 6

We have modified the sentence to make it clearer that we refer to cracking and splintering of the macroscopic solid iodine block.

Comment 7

Line 122: Here reference is made to a porous matrix. Is there a bit more information possible? It sounds like an advertising brochure again.

Response 7

The matrix is a porous aluminium oxide ceramic block with a 95% porosity designed to closely fit inside the propellant tank. It functions to both hold the iodine propellant, and improve thermal conductivity when heating the propellant. We have added this additional information on the matrix in the revised manuscript.

Comment 8

Line 134: I would like to emphasize that the more direct process $I_2 + e^- \rightarrow I^+ + I + 2e^-$ is also possible and perhaps even more likely at rather large electron temperatures (See publication by Pascaline Grondein et al. (Ref. 29)).

Response 8

We thank the referee for pointing this out, and we have modified the text to include this dissociative ionization process.

Comment 9

Line 150: How has the mass flow been calibrated? From previous publications, e.g. Ref. 30; one would expect a similar beam current at the same mass flow and rf power. The same is true for the simulation results given by Grondein et al (Ref. 29).

Response 9

The mass flow rate has been calibrated by recording the iodine tank temperature which sets the vapour pressure and sublimation rate. By precisely measuring the entire propulsion system mass before and after calibration test firings, as well as the duration of each calibration firing, the effective mass flow rate can be determined. We have added text to the revised manuscript to briefly explain the mass flow calibration.

In Ref. 30 it was found that for low mass flow rates, the power required to generate a given ion beam current was lower for iodine than xenon for the same input mass flow rate (Figure 5 in Ref. 30). Thus, if the power and mass flow rate were instead fixed, the ion beam current would be higher for iodine. Similarly, Figure 10 in Ref. 29 shows that the thruster efficiency with iodine can be significantly higher than xenon at low mass flow rates. The results from both of these references is therefore consistent with our results. We have added text in the revised manuscript noting this consistency with these previous publications.

Comment 10

Figure 2: Which RF power is given here? The power provided by the RFG or the input power of the RFG? While the former is more useful in terms of plasma physics, the latter is more important for the actual application. Some uncertainties seem to be very large, for e.g a range from about 5 % to 15 % of the mass efficiency (a factor of 3) at about 34 W and 99 $\mu\text{g/s}$. What are the reasons for this partially very large uncertainties?

Response 10

The RF power given here is the power provided by the RFG which includes the power that is finally absorbed by the plasma, as well as power losses in the antenna and matching network circuit. It does not include PPU losses, and so the actual input power is slightly higher. The total power in Figure 2 (d) however is the total power including all PPU losses. This has now been clarified in the text/caption.

The mass utilization efficiency in Figure 2 (d) is estimated based on a measurement of the total ion beam current, and the total propellant mass flow rate. The ion beam current measurement is relatively precise since it involves a measurement of the current to the grids. The propellant mass flow rate however is more uncertain since it involves measuring the change in mass of the entire propulsion system after a given firing duration. The error bars are a conservative estimate of the uncertainty largely based on the error of this mass flow rate estimate.

Comment 11

Figure 3: a) is the IEDF, not the IVDF. The IVDF should contain multiple peaks due to several m/q ratios.

Response 11

The referee raises an important point that is also quite subtle and which we discuss more fully in the response to Comment 20 below.

Comment 12

Line 358: what exactly means global mass? And what is a global continuity?

Response 12

Global mass refers to a statement of mass conservation for the thruster. Here, the input propellant mass flow rate must be equal to the sum of the ion beam mass flow rate and the remaining un-ionized neutral gas mass flow rate leaving the thruster at steady-state conditions.

Global continuity refers to a statement of particle conservation within the plasma discharge. Here, at steady-state conditions, the addition of new plasma due to ionization must balance the loss of plasma at the source tube radial walls, back wall, screen grid, and leaving the thruster. The global continuity equation is obtained from the integral form of the usual differential continuity equation with a control volume surrounding the ICP plasma, and with a control volume boundary located at the sheath edge (for example). We have clarified the text to address some of the above points.

Comment 13

Line 369: Is a mean mass used to calculate the Bohm velocity or is the velocity calculated for each ion species? The used approach may influence the results quite significantly.

Response 13

In the results presented in our manuscript, only xenon is used in the plasma model, which we have now more clearly stated in the revised manuscript. Thus, the mass of atomic xenon is used to calculate the Bohm velocity. Xenon was used extensively in the early design and development of the propulsion system and an iodine model with detailed collisional processes is currently being developed.

Comment 14

Line 407: Which ion mass was assumed for calculating the perveance? Since the perveance is proportional to the inverse square root of the ions mass, this should at least be mentioned.

Response 14

We thank the referee for pointing out this omission in our manuscript. We have used the atomic iodine mass to calculate the perveance. This has now been mentioned in the revised manuscript.

Comment 15

Line 416: This factor should be introduced directly with the Eq. in line 176. Perhaps it should be mentioned that the pre-factors simply represent the root of the mass-charge ratio to the iodine atom.

Response 15

This factor has now been introduced earlier in the manuscript in line 176, and we have clarified the pre-factors. The thrust correction factor section in Methods has been removed.

Comment 16

Line 470: The actual required heating power may be much larger due to radiative losses compared to the power required for sublimation. This should be discussed at least by an example.

Response 16

Because of the reuse of waste heat from the thruster and power electronics, less than 1 W of additional heating power is needed to maintain the tank at the required operating temperatures during steady-state operation. This has now been mentioned in the revised manuscript.

Comment 17

Lines 502-506: What is estimated pumping speed for atomic and molecular iodine? The correction factor for iodine may suffer large uncertainties, since the composition (I and I₂) of the neutral gas close to the pressure gauge may not be known well. How was this issue solved?

Response 17

As the referee notes, it is difficult to accurately determine the gas composition near the pressure gauge. However, during operation the pressure can be measured with a capacitance gauge (with a resolution of 0.01 mTorr) which is independent of the gas composition. For lower pressures, a cold cathode Penning pressure gauge was used, with gas-correction factors estimated by comparison with the capacitance gauge at slightly higher pressures. The pumping speed still depends on the effective mass of the gas though, and so we are only able to estimate lower and upper bounds with an actual pumping speed between about 700-1400 l/s. This estimated pumping speed has been added to the revised manuscript.

Comment 18

Line 513: Have negative iodine ions been detected in the beam? Negative iodine ions may not be able to overcome the sheath potential and should mainly remain in the discharge until recombination. Or is there a high amount of electron-capture processes between neutral iodine and electrons?

Response 18

Inside the ICP discharge any possible negative ions would not of course be able to escape. However, due to the presence of low-temperature electrons in the plume produced from the cathode, we cannot rule out the possibility that some negative ions may be present in the plume. Some initial experiments have indicated the presence of negative ions, but due to the many different collisional processes, and the low plasma density, further work is needed to fully understand the plume physics.

Comment 19

Lines 527-537: What is the distance between thruster (grid) and molybdenum disk? Normally, a TOF measurement requires a fast stop signal. A simple current probe seems to me being a slow device with high time constants. Is there any additional/supporting technique used to increase quality of the TOF spectra? What is the rise and fall time of the high voltage pulse?

Response 19

The distance between the thruster and the molybdenum disk is approximately 54 cm. Although the probe is relatively simple, the transit time of ions to the probe is much higher than the estimated time constant. This time constant is also reduced by using short low impedance connections, and is estimated to be of the order of 1 μ s. The rise and fall time of the high-voltage pulse is approximately 0.5 μ s. This information has now been added to the revised manuscript. The investigation was also only interested in the main iodine species (I^+ , I_2^+ , I_2^{2+}), which allowed us to limit the measured m/z spectrum.

Comment 20

Lines 552-554: It is claimed that a retarding field energy analyzer measures the velocity function and not the energy distribution function. That is not correct. Publications on RFEAs consistently claim that the IEDF is measured (see for instance Gahan et al. "Retarding field analyser for ion energy distribution measurements at a radio-frequency biased electrode", Review of Scientific Instruments 79, 033502 (2008); <https://doi.org/10.1063/1.2890100>). And anyway, what is a velocity supposed to be in energy units? A velocity (or momentum) filter normally requires a magnetic field. Furthermore, to be more concrete: the RFEA determines the distribution function of the accelerating potential.

Response 20

The distribution obtained from an RFEA is very subtle and requires a careful discussion. Before treating the RFEA, we define what we mean by velocity and energy distributions. Considering for sake of argument a 1D Maxwell-Boltzmann distribution of a single ion species, the velocity distribution function (VDF), $f(v)$, is

$$f(v) = B e^{-\frac{mv^2}{2k_B T}}$$

where B is a constant, m is the mass, T is the temperature, and k_B is Boltzmann's constant. This equation is defined in such a way that: $n = \int_{-\infty}^{\infty} f(v) dv$, where n is the density. The equation for the VDF is in velocity units, or on a velocity scale. By defining the energy as $\varepsilon = mv^2/2$, this can be written in energy units, or on an energy scale as

$$f\left(\sqrt{\frac{2\varepsilon}{m}}\right) = B e^{-\frac{\varepsilon}{k_B T}}$$

Despite this quantity being written in terms of the energy, it is not the Energy Distribution Function (EDF), and is sometimes referred to as the Energy Probability Function (EPF). To obtain the true EDF, $g(\varepsilon)$, the 1D Jacobian of the velocity-to-energy transformation must be considered, $f(v)dv \rightarrow g(\varepsilon)d\varepsilon$, which gives

$$g(\varepsilon) = \frac{f(v)}{mv} = \frac{1}{\sqrt{2m\varepsilon}} f\left(\sqrt{\frac{2\varepsilon}{m}}\right)$$

Defined in this way one obtains: $n = \int_0^{\infty} g(\varepsilon)d\varepsilon$. RFEA's nominally measure the current as a function of discriminator voltage to obtain a current-voltage characteristic. Ignoring the transparency of the RFEA grids and again considering only a single ion species, the current measured by an RFEA is

$$I = qA \int_{v_{min}}^{\infty} v f(v) dv$$

where q is the ion charge, A is the probe collecting area, and $v_{min} = \sqrt{\frac{2q\phi_d}{m}}$ with ϕ_d the probe discriminating voltage. Taking the derivative of the probe current with respect to this voltage gives

$$\frac{dI}{d\phi_d} = -\frac{q^2 A}{m} f\left(\sqrt{\frac{2q\phi_d}{m}}\right)$$

Thus, the derivative is proportional to the VDF in energy units, not the EDF. This derivative method is what we have used to obtain our distributions, and is typically the most common analysis method used with RFEA's. By contrast, to find the EDF, the 1D velocity-to-energy Jacobian transformation must again be considered. In the paper by Gahan *et al*, it is not completely clear what method is used by those authors.

In the case of multiple ion species, a similar analysis shows that the derivative of the RFEA current-voltage characteristic is still related to the VDF of the different species. Because the ions are accelerated through the same potential, and since the VDF is in energy units, there is only a single peak in the RFEA distribution and different q/m ratios cannot be determined. To obtain the VDF in velocity units one would also need to consider the 1D Jacobian transformation, but in the presence of multiple ion species this is challenging without additional information since the RFEA only discriminates energy, not velocity.

In the context of capacitively coupled plasmas, such as those used in the semiconductor industry, multiple ion species are often encountered, and the ion distributions measured on the RF electrodes are often referred to as the Ion Flux Distribution Function (IFDF), $h(\varepsilon)$, defined such that

$$\Gamma = \int_0^{\infty} h(\varepsilon) d\varepsilon$$

where Γ is the flux. Although this looks similar to how the EDF is defined, the quantity $h(\varepsilon)$ is not the EDF since its integral gives a flux, not a density, and from the discussion above, it can again be related to the VDF in energy units.

Since the discussion above is a very subtle issue, and to avoid confusion to readers, we have now rather used the label “IFDF”, and explicitly mentioned in the manuscript how we define this distribution.

Comment 21

One last point on axis labelling for graphs. I am not aware of any further specifications from Nature here and in my opinion it is not my task either, but for the sake of completeness I want to point out that units in square brackets is the only notation that is rejected in all valid standards (e.g. in ISO 31-0). I would recommend a standard-compliant variant here.

Response 21

To be consistent with other Nature papers, we have replaced square brackets with round brackets for the units in all graphs of our revised manuscript.

Reviewer Reports on the First Revision:

Referees' comments:

In light of their opinion of this article, which hasn't changed, Reviewer #1 decided not to participate in the peer-review of this work further.

Referee #2 (Remarks to the Author):

The paper "First in-orbit demonstration of an iodine electric propulsion system," by Rafalskyi et al., describes a first flight of an iodine-fed electric propulsion system. The paper discusses the thruster design and diagnostics used to quantify the thruster's performance on the ground and in flight, with comparisons between the two to show the efficacy of the propulsion system.

The comments offered on the original manuscript by this referee were minor and well-addressed by the authors in their revised manuscript. I also thought that the authors did a good job of addressing the points raised by the other referees. I have no further comments to offer at this time, and recommend this paper for acceptance and publication.

Referee #3 (Remarks to the Author):

Since this is a revised version of the article "In-orbit demonstration of an iodine electric propulsion system" by Lafleur and coworkers, I will be brief here: The authors have answered my points from the first report satisfactorily and updated corresponding passages in the manuscript. So I have no concerns here that would stand in the way of publication. There is one thing I would like to emphasise here, as one of the other reviewers doubts the novelty of an iodine engine. In a way, I can understand the reviewer's concerns. Iodine has been discussed as a propellant for almost two decades now and there are already some prototypes in laboratories. Ion engines are also a well-known technology that was first used in space in the 1960s. However, the first successful use of an iodine-powered ion thruster in space must be considered separately. I think this should outweigh the concerns.

Consequently, my basic recommendation to publish the article in Nature has not changed. The results shown appear valid and consistent. The text is written in a comprehensible way so that even non-specialist readers can understand it.